# Mutational scanning reveals oncogenic *CTNNB1* mutations have diverse effects on signaling

Anagha Krishna[1], Alison Meynert [2], Karamjit Singh Dolt [3], Martijn Kelder[2], Agavni Mesropian [4,5], Ailith Ewing [2], Conny Brouwers[6], Jill WC Claassens[6], Margot M. Linssen [6], Shahida Sheraz[1], Gillian CA Taylor[2], Philippe Gautier[2], Anna Ferrer-Vaquer[7], Graeme Grimes[2], Hannes Becher[2], Ryan Silk [2], Albert Gris-Oliver [4,5], Roser Pinyol[4,5], Colin A. Semple[2], Timothy J. Kendall [2], Thomas Graham Bird [8,9,10,11], Anna-Katerina Hadjantonakis[7], Joseph A. Marsh [2], Josep M. Llovet[4,5,12,13], Peter Hohenstein [1,3,6] ✉, Andrew J. Wood [2] ✉ & Derya D. Ozdemir [1,14,15] ✉

*CTNNB1*, the gene encoding β-catenin, is a frequent target for oncogenic mutations activating the canonical Wnt signaling pathway, typically through missense mutations within a degron hotspot motif in exon 3. Here, we combine saturation genome editing with a fluorescent reporter assay to quantify signaling phenotypes for all 342 possible missense mutations in the mutation hotspot. Our data define the genetic requirements for β-catenin degron function, refine the consensus motif for substrate recognition by β-TRCP and reveal diverse levels of signal activation among known driver mutations. Tumorigenesis in different human tissues involves selection for *CTNNB1* mutations spanning distinct ranges of predicted activity. In hepatocellular carcinoma, mutation effect scores distinguish two tumor subclasses with different levels of β-catenin signaling, and weaker mutations predict greater immune cell infiltration in the tumor microenvironment. Our work provides a resource to understand mutational diversity within a pan-cancer mutation hotspot, with potential implications for targeted therapy.

The canonical Wnt pathway is essential for normal development and is erroneously activated in many cancers[1]. Its main intracellular effector, β-catenin (encoded by *CTNNB1*), is normally degraded by a destruction complex consisting of APC, AXIN2, CK1 and GSK3β. Wnt ligand binding or mutations that impair this complex lead to β-catenin accumulation, nuclear translocation and activation of T cell Factor (TCF)/lymphoid enhancer-binding factor (LEF) transcription factor target genes (Supplementary Fig. 1a). The tumor suppressor adenomatous polyposis coli (APC) serves as a scaffold within the destruction complex. *APC* loss-of-function mutations activate β-catenin and are common in colorectal cancers, in which tumors often select for mutations with a 'just-right' level of β-catenin signaling[2–5].

Gain-of-function changes within the *CTNNB1* exon 3 degron motif are among the most common mutations in several tumor types[6,7]. These mutations act in a dominant manner by disrupting the interaction between the phosphorylated DpSGXXpS motif (residues 32–37) and the E3 ubiquitin ligase receptor β-TRCP, preventing β-catenin ubiquitylation and degradation[7]. The phospho-regulatory cascade involves CK1 priming at S45, followed by GSK3β phosphorylation at T41, S37 and S33, enabling β-TRCP docking.

---

Mutations within exon 3 of *CTNNB1* occur in 20–30% of patients with hepatocellular carcinoma (HCC)[8,9]. HCC has been classified into subtypes with distinct molecular and immune features[10–13]. The inflamed class is characterized by interferon signaling, enrichment of cytolytic T cells and M1 macrophages, whereas the non-inflamed class is largely devoid of immune infiltrate. β-catenin signaling and *CTNNB1* mutations are associated with immune exclusion and promote resistance to immunotherapy in mouse models of HCC[9,14–16]. However, a substantial fraction of *CTNNB1*-mutant HCCs presents an inflamed phenotype (30%), highlighting the need for studies to elucidate the underlying mechanisms[14].

Previous studies have suggested that different *CTNNB1* mutations can activate signaling to different degrees[17–19]. A small subset of exon 3 genotypes observed in human cancer has been directly compared using cellular assays; however, the functional impact of most variants remains unknown. Here, we develop a saturation genome editing (SGE) assay to systemically quantify the relative activity of all possible missense substitutions within the exon 3 hotspot, assigning functional scores to >80% of *CTNNB1* missense mutations observed in cancer.

## Results

### *CTNNB1* hotspot mutational patterns are tissue-specific

We analyzed 9,248 tumors with *CTNNB1* mutations in the COSMIC database (Fig. 1a, Supplementary Fig. 1b and Supplementary Table 1). As previously reported[6], the majority of mutations (86%) were missense substitutions. Of these, 88% occurred in the L31–G48 hotspot region encoded by exon 3 (Fig. 1a,b and Supplementary Table 1). Residues T41 and S45 were most frequently affected, comprising 27% and 25% of hotspot missense mutations, respectively, followed by S37, S33, D32 and G34 (Supplementary Fig. 1b).

Analysis across tissues revealed tissue-specific mutation preferences (Fig. 1c and Supplementary Fig. 2). S45 mutations are greatly enriched in adrenal and kidney tumors, whereas mutations within the β-TCRP docking motif predominated in the central nervous system (Fig. 1c and Supplementary Fig. 2). Liver and skin showed a broader profile, encompassing all six of the most frequently mutated positions (Fig. 1c and Supplementary Fig. 2). Specific amino acid substitutions at each residue also varied by tissue. For example, 99% of soft tissue tumors with T41 mutations (1,201 out of 1,217) carried T41A, whereas in pituitary gland, 89% of T41 mutations (77 out of 87) were T41I (Supplementary Fig. 2). Overall, *CTNNB1* hotspot mutational patterns are highly diverse and tissue-dependent.

### Mutational scanning reveals diverse consequences of *Ctnnb1* hotspot mutations

To systematically measure the consequences of *CTNNB1* hotspot mutations on β-catenin signaling, we devised a multiplexed CRISPR homology-directed repair (HDR) assay covering all 342 possible single amino acid substitutions across positions 31–48 (Fig. 1d). This assay enables phenotypic analysis of variants expressed from the endogenous mouse *Ctnnb1* locus, which is 100% conserved with human exon 3 at the amino acid level. Variant function was measured in parallel under normal regulatory control[20–22], providing a sensitive and comprehensive assessment of hotspot mutant phenotypes. A library of HDR templates was synthesized, each encoding a single amino acid substitution and flanking synonymous changes to enable selective PCR amplification of edited alleles[20,21] (Fig. 1d).

To analyze individual hotspot mutations at the single-cell level, we derived embryonic stem (ES) cells from the Tcf/Lef-H2B–GFP transgenic mouse line[23,24], then replaced exons 2–6 of *Ctnnb1* with a negative counter-selection cassette on one allele (Fig. 1d). Specifically targeting and replacing the selection cassette with a multiplexed HDR template library, combined with fluorescence-activated cell sorting (Supplementary Fig. 3a), enabled efficient quantification of the functional output of each edited allele (Fig. 1d). ES cells were chosen

for their high HDR efficiency and intact canonical WNT signaling, in contrast to most tumor lines.

After editing and selection, a subset of cells displayed elevated Tcf/Lef-H2B–GFP reporter expression (Supplementary Fig. 3a), consistent with canonical Wnt pathway activation. Cells were sorted into six equally log-spaced bins (P1–P6) based on increasing reporter GFP signal (Supplementary Fig. 3a), and genomic DNA was subjected to amplicon deep sequencing across the *Ctnnb1* hotspot region using primers specific for HDR-edited alleles (Fig. 1d). Amplicon sequencing was also performed on the untransfected HDR donor library ('plasmid') and edited but unsorted cells ('pool'; Supplementary Fig. 3b), enabling quantification of *Ctnnb1* variant frequencies across reporter activity bins relative to their frequency in the total cell population (Fig. 1e). Normalized mutant allele frequencies showed high correlations between biological replicates (Pearson's $r$, range 0.54–0.89; Supplementary Fig. 3c) and were merged for downstream analysis. Frequencies also correlated well between plasmid and pool samples (Pearson's $r$, 0.63), indicating consistent and unbiased HDR efficiency throughout the targeted region (Supplementary Fig. 3c).

The mutation frequency was relatively constant across positions in the unsorted cell population (Supplementary Fig. 3b), whereas it varied substantially across cells with different GFP levels (Fig. 1e). Substitutions at codon positions that are frequently mutated in tumors (D32, S33, G34, S37, T41 and S45) were rarely observed in low activity bins but frequently observed in bins of higher activity (Fig. 1e), validating this system's ability to functionally classify *CTNNB1* mutations.

### Calculation and validation of mutation effect scores

For each variant, we calculated a mutational effect score (MES) by integrating allele frequencies and reporter activation levels across bins[25] (Supplementary Methods). This procedure provided a metric to compare the phenotypic consequences of each substitution and gain insight into β-catenin regulation (Fig. 2a and Supplementary Table 2).

We first compared the MES values with scores produced by 50 computational variant effect predictors (VEPs) for the same substitutions[26] (Supplementary Table 3). The strongest correlation was observed for EVE, which uses deep generative models to calculate variant effect scores from evolutionary sequence information[27]. The correlation between EVE and MES scores for the *CTNNB1* hotspot (Spearman's $\rho$, 0.662) was higher than any other EVE–deep mutational scanning (DMS) combination and all but three of 1,430 VEP–DMS comparisons in a prior benchmarking study[26]. Although correlation with VEP scores is not a definitive measure of DMS reliability, the strong concordance supports the high quality of the current dataset.

We next tested the ability of MES scores to predict endogenous β-catenin target activation. Using multiplex-ready ES cells, we generated 15 clonal cell lines (Fig. 2b), each heterozygous for one of 11 mutations spanning a range of MES values, focusing on substitutions at the three most frequently mutated positions in human cancer. Transcriptional responses were measured using bulk 3′ RNA sequencing (RNA-seq), with GSK3β inhibition by CHIR99021 serving as a positive control. Unsupervised hierarchical clustering classified cell lines into low, medium and high MES categories (Fig. 2c). Ten β-catenin targets that showed dose-responsive activation by CHIR99021 (Pearson $r > 0.85$; Fig. 2d) also displayed positive correlations between transcript levels and the relevant MES value (Fig. 2e; median Pearson $r = 0.57$; range, 0.02–0.87; $P < 0.05$ for seven out of ten target genes).

MES values also predicted endogenous β-catenin target gene expression in human fetal liver organoids. A previous study introduced one of four exon 3 mutations (D32G, S33F, T41A or S45P) homozygously at the endogenous *CTNNB1* locus (Supplementary Fig. 4a)[28]. RNA-seq data from these organoids showed positive correlations between MES values and β-catenin target genes in nine out of ten cases (median Pearson $r = 0.65$; range, −0.5 to 0.96; Supplementary Fig. 4b). As outlined further below, MES scores also predicted the strength of

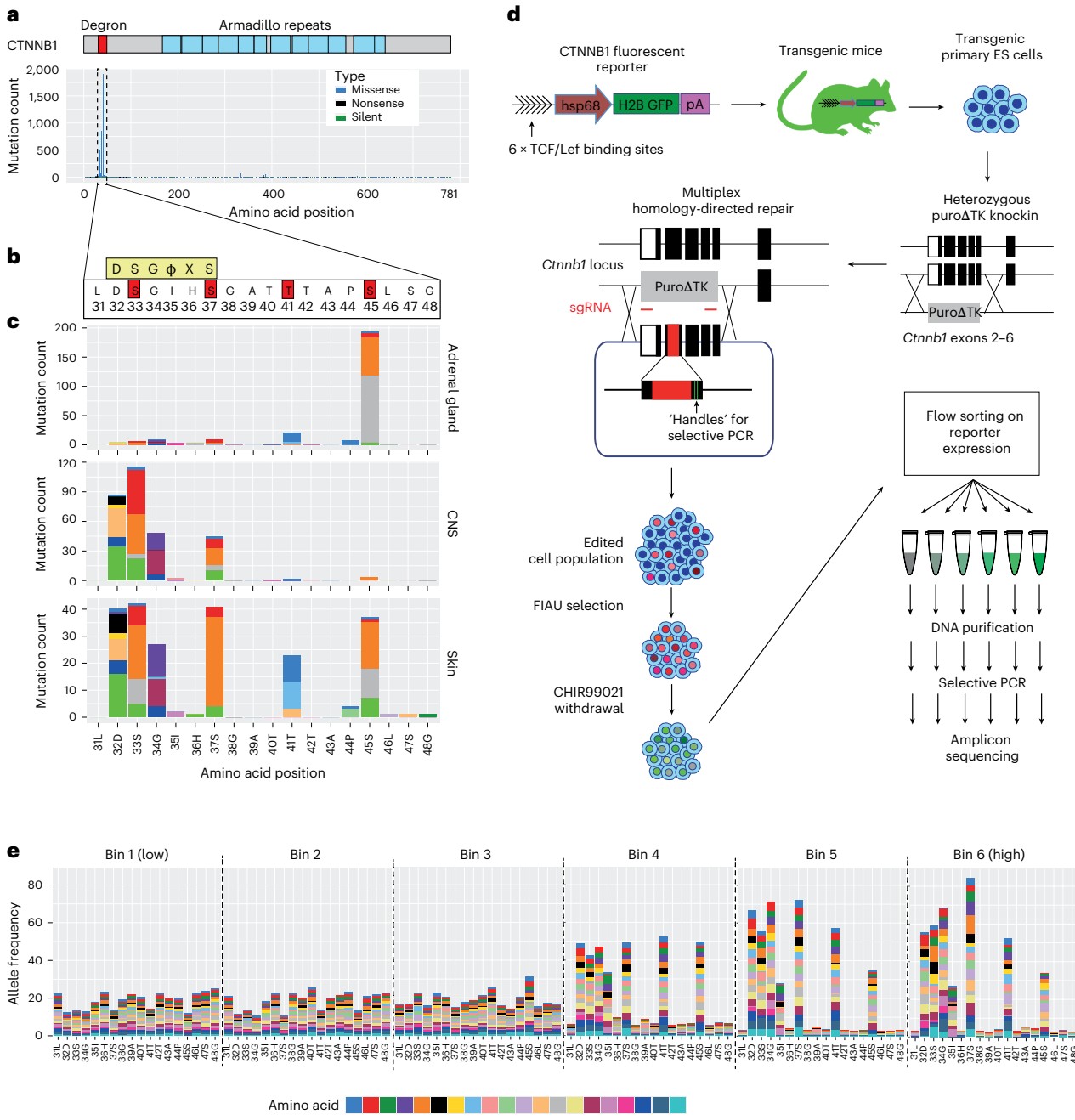

**Fig. 1 | Tissue-specific mutation profiles at the *CTNNB1* degron in human cancer. a**, Histogram showing the frequency of mutations at each amino acid position of human *CTNNB1* across all tumors present in the COSMIC database. Blue boxes represent the positions of armadillo repeat domains, and the red box indicates the degron and mutation hotspot, which is expanded in **c**. **b**, The amino acid sequence of the *CTNNB1* mutation hotspot. The consensus docking site for the β-TRCP E3 ligase substrate receptor is shown in a yellow box; φ indicates any hydrophobic amino acid, and X indicates any amino acid. Phosphorylation sites known to be critical for β-catenin turnover are highlighted in red boxes. **c**, Histograms show distinct distributions of mutations within the *CTNNB1* mutation hotspot for COSMIC tumors of the adrenal gland, central nervous system (CNS) and skin. Data from other primary tissue sites are shown in Supplementary Fig. 2. **d**, Schematic overview of the mutational scanning assay. A β-catenin signaling activity reporter construct was integrated randomly into the genome to generate transgenic mice[23]. Primary ES cells were derived, and a region spanning exons 2–6 of *Ctnnb1* was replaced with a puroΔTK selection cassette on one of two alleles. A plasmid library was generated for homology-directed repair, encoding each of 342 possible single amino acid

substitutions spanning codons 31 to 48 of β-catenin, together with silent mutations to allow selective PCR amplification of edited alleles from genomic DNA. This was transfected into multiplex-ready mouse ES cells together with sgRNAs that cut in an allele-specific manner on either side of the selection cassette. After recovery, transfected cells were cultured in the presence of FIAU to kill those in which the selection cassette had not been removed. Multiplex-ready cells were routinely cultured under 2i conditions[51] to compensate for hemizygous *CTNNB1* expression, but the GSK3β inhibitor (CHIR99021) was withdrawn 2 days before sorting to allow β-catenin signaling to return to a baseline state. Cells were then subject to fluorescence-activated cell sorting based on the level of GFP reporter expression, before extraction of genomic DNA, amplification of *Ctnnb1* exon 3 by PCR and Illumina sequencing. Further methodological information is detailed in the Methods. **e**, The frequency of individual missense mutations across each position in the mutation hotspot in cell populations sorted according to the scheme shown in **d**, expressed relative to their frequency in the unsorted pool sample. The color scheme used to represent different missense mutations is shown to the right.

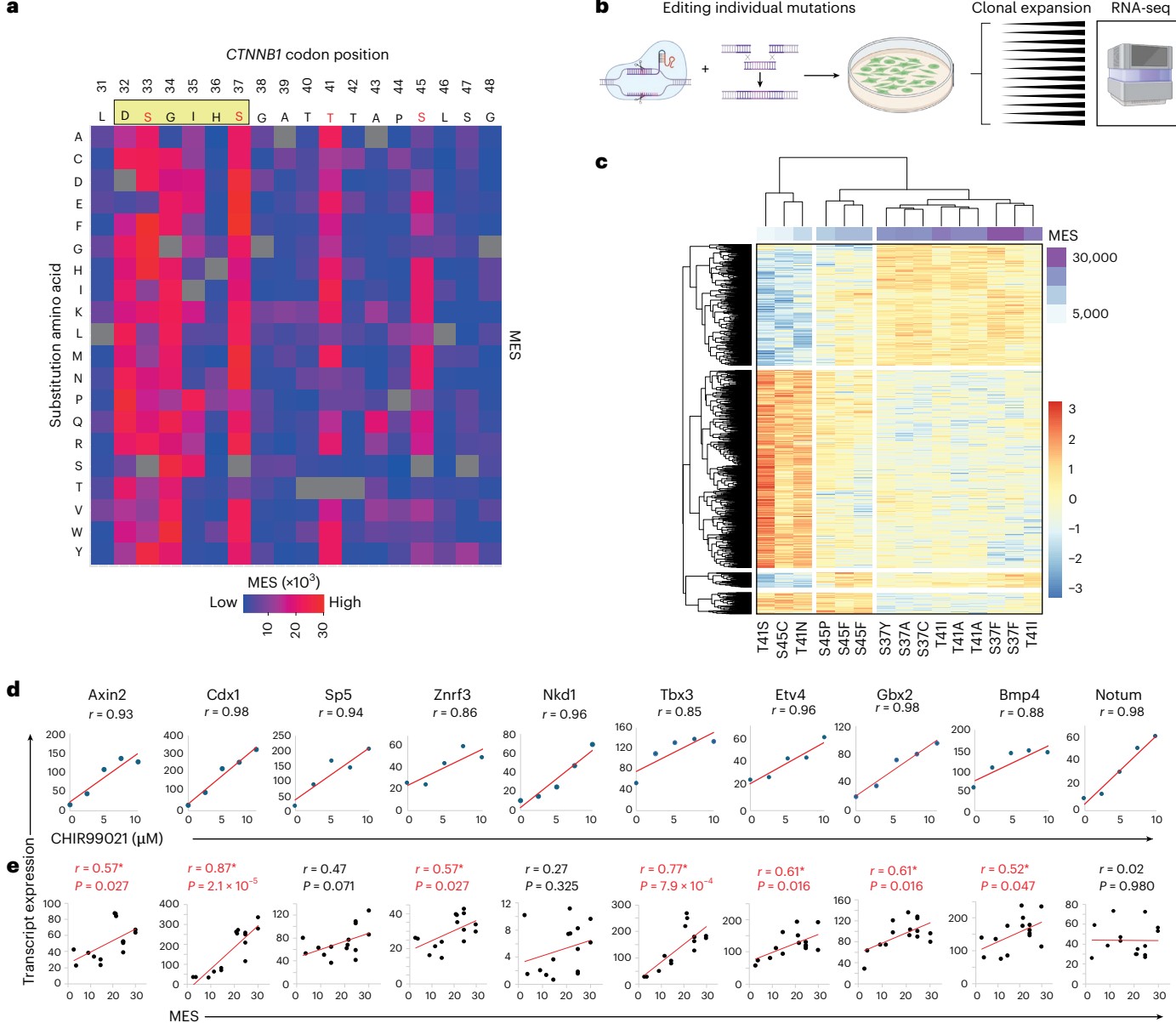

**Fig. 2 | MESs predict endogenous β-catenin target gene expression. a**, Heatmap representation of MESs, showing the activity profile for every possible amino acid substitution at the *CTNNB1* mutation hotspot, reconstructed from data shown in Fig. 1e. Individual scores are provided in Supplementary Table 2. The consensus motif for β-TRCP docking is shown in a beige rectangle above the heatmap, with known phosphorylation sites highlighted in red. **b**, Schematic to illustrate the derivation of clonal mouse ES cell lines genome-edited to express individual *Ctnnb1* exon 3 mutations. Created in BioRender.com. **c**, Unsupervised clustering of *Ctnnb*-mutant clones based on global RNA-seq expression profiling shows co-clustering of clones with mutations of similar MES values (purple bar). **d,e**, Ten endogenous β-catenin target genes were selected based on the high correlation (>0.85) between transcript expression and dose of the GSK3β inhibitor CHIR99021 (**d**); transcript expression also correlated with MES values of the relevant exon 3 mutations (**e**). All *r* values and *P* values are from Pearson tests (two-sided); *P* values of <0.05 are highlighted in red.

β-catenin target activation in two independent human HCC cohorts. Altogether, these data show that MES values predict the degree to which endogenous β-catenin target genes are activated by different exon 3 mutations in both mouse ES cells and hepatocytes.

**Refining genotype–phenotype correlations within the *CTNNB1* hotspot**

The current model of genotype–phenotype relationships within the exon 3 hotspot proposes that missense mutations exert position-dependent effects on β-catenin signaling: S45 mutations activate weakly, T41 mutations activate moderately and mutations within the core β-TRCP docking motif (positions 32–37) activate strongly[19].

Our derivation of MES values for all 19 possible missense mutations across the hotspot region enabled testing and refinement of this model. Notably, of the 342 possible amino acid changes, only 105 can arise by a single nucleotide substitution (median of six changes per site; Fig. 3a). As expected, this group accounts for >98% of the single amino acid substitutions in the COSMIC database (Supplementary Table 1).

Consistent with the current model, we found that S45 mutations generally produced weaker activation than mutations in the docking motif, both among single-nucleotide variants and across all substitutions (Fig. 3b). Nonetheless, MES values at the same position varied substantially (Fig. 3a,c). For example, S45T and T41S were well tolerated (Fig. 3c), reflecting the ability of CK1α and GSK3β to phosphorylate

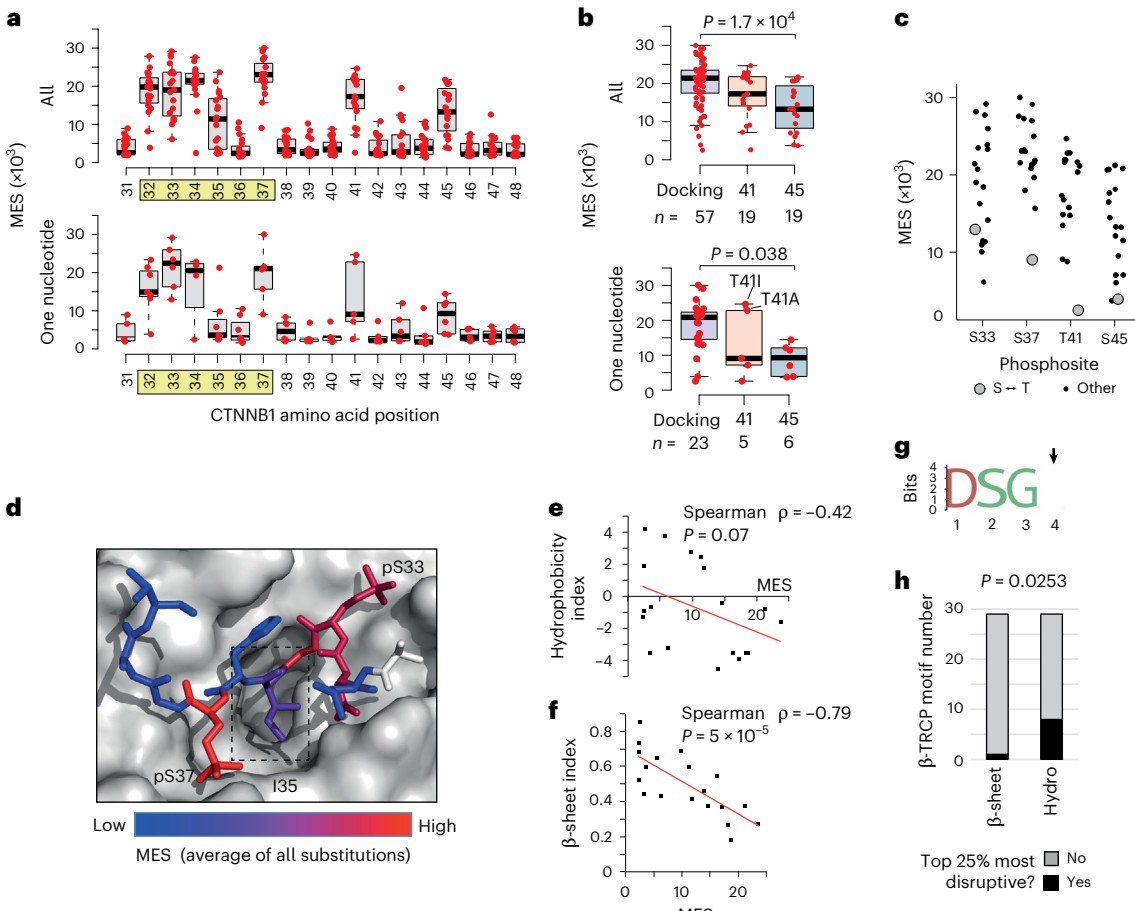

**Fig. 3 | Genotype–phenotype correlations at the *CTNNB1* mutation hotspot.**
**a**, The distribution of MES values is shown by codon position for all 19 missense substitutions (top) or just the subset that can be reached by a single nucleotide mutation in the human genome (bottom). The core β-TRCP docking motif is highlighted in yellow. For boxplots in **a** and **b**, horizontal lines show the median value, boxes show the second and third quartiles and whiskers show the range. **b**, The distribution of all MES values (top) or just the single nucleotide subset (bottom) for invariable positions within the β-TRCP docking motif (combined for D32, S33, G34, S37: 'Docking'), T41 and the CK1 target site at position 45. The two most common T41 mutations in human cancer are labeled in the bottom panel. *P* values show one-way ANOVA with post hoc Tukey's honestly significant difference test. **c**, The distribution of MES values for all amino acid substitutions is shown for individual phosphosites, with serine–threonine substitutions highlighted. **d**, Structure of human β-TRCP (shown in surface mode) in complex with the phosphorylated degron peptide of β-catenin (amino acids 30–40 are shown) based on PDB 1P22. Individual β-catenin residues are colored according to the mean MES value across all substitutions at that position. The box indicates position I35. **e**, Correlation between MES values for individual substitutions at position I35 with the Kyte–Doolittle hydrophobicity scale[34]. **f**, Correlation between MES values for individual substitutions at position I35 with the Chou–Fasman β-sheet scale[36]. *P* values in **e** and **f** show Spearman's rank tests (two-tailed). **g**, Amino acid logo generated from *n* = 29 high-confidence β-TRCP docking sites containing the 'DSGX' motif[37]. Position 4, equivalent to I35 in β-catenin, is highlighted with an arrow. **h**, Stacked bar chart showing the proportion of high-confidence β-TRCP motifs (*n* = 29), with position 4 residues that rank in the top 25% most disruptive based on the hydrophobicity and β-sheet indices detailed above. *P* value shows Fisher's exact test (two-tailed).

either serine or threonine[29]. Substitution of S45 for small amino acids (alanine, glycine) was also well tolerated (Fig. 2a). It has previously been reported that in-frame deletions at S45 still permit phosphorylation at T41, S37 and S33 in colon cancer cells[30,31]. Collectively, the data thus suggest that the absence of a large side chain at S45 enables β-catenin degron function without CK1α priming[32].

Although the average effect of all T41 substitutions was intermediate, >98% in the COSMIC database are substitutions for alanine or isoleucine, which were the two strongest activating substitutions identified at this position. Both substitutions can be reached by one nucleotide change (Fig. 3c). This contrasts with S45, for which all eight strongest amino acid substitutions require at least two nucleotide changes (Fig. 3c). Therefore, strong activating S45 mutations are theoretically possible (for example, S45M, S45K) but are rarely observed in cancer, probably because they require at least two nucleotide changes.

Extending the current model[19], our data show that T41A and T41I activate β-catenin more than many commonly observed docking-motif variants (for example, D32V, D32Y, D32H, S33P, G34R). Other mutations at T41 (T41N, T41P), which are rare but still recurrent in human cancer (Supplementary Table 1), elicit weaker activation in the lower range of S45 mutations. It is therefore evident that different substitutions at the same position can elicit markedly different effects on β-catenin signaling, refining our understanding of *CTNNB1* mutational diversity.

I35 lies at the center of the docking motif, directly contacting β-TRCP, and is thought to require a hydrophobic amino acid side chain[6,33]. The I35 side chain projects into a cavity on the β-TRCP surface[33] (Fig. 3d). MES values for I35 substitutions were broadly distributed (Figs. 2a and 3a) and negatively correlated with a hydrophobicity index at this position[34] (Spearman ρ = −0.42; Fig. 3e). However, substitution of isoleucine for several polar amino acids (for example, threonine, asparagine, tyrosine) yielded low MES values (Fig. 2a), indicating that hydrophobicity is not essential. By screening 566 amino acid property indices[35], we observed that the strongest correlations with I35 MES values were nearly all related to secondary structure propensities.

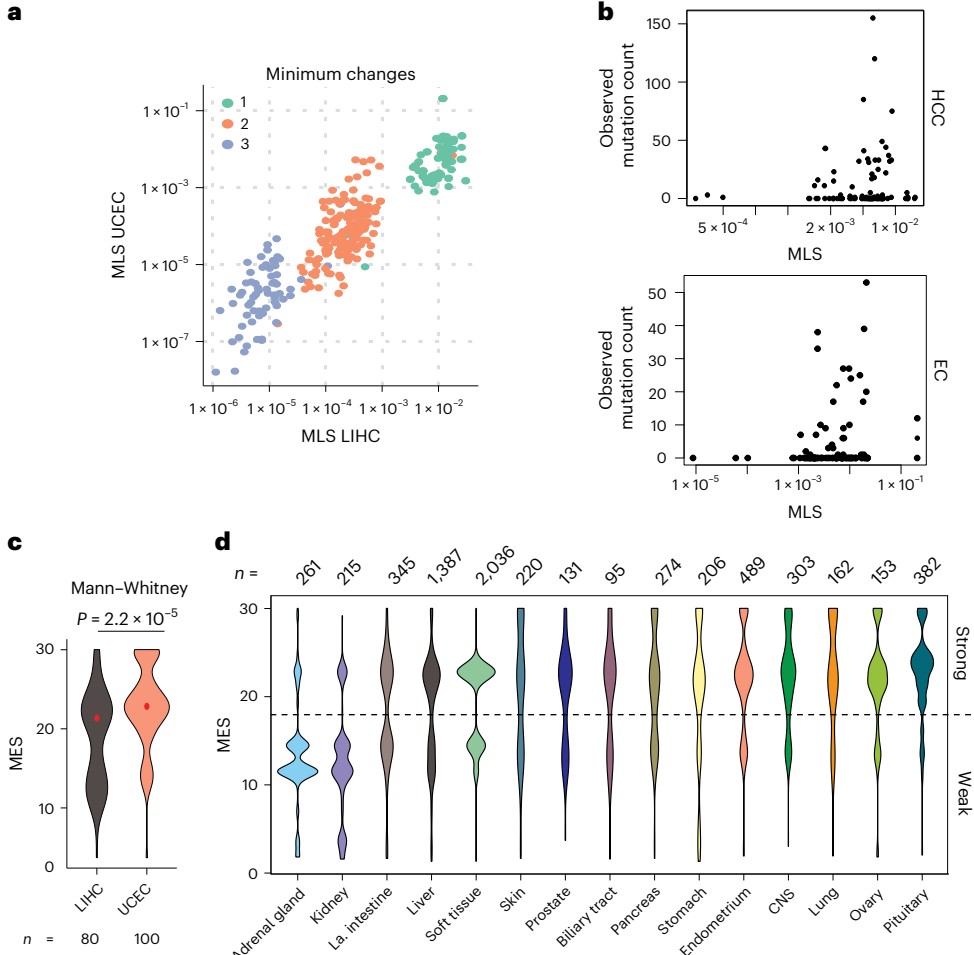

**Fig. 4 | Tissue-specific *CTNNB1* mutation patterns are driven by selection for optimal levels of β-catenin signaling. a**, Relationship between MLSs (see Supplementary Fig. 5) in exomes of HCC (LIHC) and endometrial carcinoma (UCEC) tumors from TCGA. Scores are shown for all 342 possible amino acid changes, colored according to the minimum number of nucleotide substitutions required. **b**, Relationship between the observed frequency of specific *CTNNB1* hotspot amino acid substitutions in the COSMIC database and MLSs calculated from LIHC and UCEC exomes from TCGA (see Supplementary Fig. 5). Only amino acid substitutions that can be reached by a single nucleotide mutation (>98% of observed mutations) are shown. EC, endometrial carcinoma. **c**, Violin plots show the distribution of MES values of *CTNNB1* exon 3 hotspot mutations in LIHC and UCEC tumors from TCGA. *P* value shows two-tailed Mann–Whitney test without adjustment for MLS. **d**, The distribution of MES values in tumors from different primary sites in the COSMIC dataset. All tissue sites with >100 *CTNNB1* exon 3 mutations are shown. Tissues are ordered according to the ratio of weak to strong hotspot mutations.

A Spearman correlation of −0.79 was observed with the 'normalized frequency of β-sheet' index[36] (Fig. 3f). Although residue I35 does not form a β-sheet in any available crystal structure, its dihedral angles place it clearly in the β region of a Ramachandran plot.

To ask whether this secondary structure requirement extended to other β-TRCP docking sites, we analyzed 28 high-confidence β-TRCP-dependent degrons containing the 'DSGX' motif[37] (Supplementary Table 4). Position 4 residues in this motif (corresponding to I35 in β-catenin) were variable across substrates (Fig. 3g). Numerous substrates (eight out of 28) had position 4 amino acids that ranked in the 25% most disruptive on the hydrophobicity scale, whereas only one out of 28 featured in the same bracket of the β-sheet scale (Fisher's exact test, *P* = 0.0248). Therefore, an extended backbone conformation, rather than side chain hydrophobicity, better explains the effects of I35 substitutions and potentially analogous positions in other β-TRCP docking motifs.

## Tissue-specific *CTNNB1* mutation patterns are driven by selection for optimal levels of β-catenin signaling

The frequency of *CTNNB1* hotspot mutations varies across tumor types (Fig. 1c and Supplementary Fig. 1), probably reflecting multiple contributing factors. Stem cells in different tissues experience different genotoxic insults, which affect overall mutational spectra[38,39] as well as the probability that specific *CTNNB1* missense mutations become available for selection. Alternatively, or in addition, different tissue environments might favor selection for missense mutations causing levels of activation that are optimal, or 'just-right', for their spatio-temporal context[2]. The current data provide an opportunity to distinguish these possibilities.

Mutational probability, calculated from background nucleotide substitution rates, is a poor predictor of *CTNNB1* mutation patterns in cancer[40], suggesting a strong influence of selection. We computed 'mutational likelihood scores' (MLSs) for each of 342 hotspot missense mutations, using the background rates of nucleotide substitution in HCC and endometrial carcinoma whole-exome sequencing data (Supplementary Fig. 5a,b). These scores represent the probability of an amino acid substitution, given the dominant mutational biases seen genome-wide in coding sequences of *CTNNB1*-mutant tumors. MLS values correlated positively across tissues (Pearson *r* = 0.920 for all mutations, *r* = 0.655 for the single-nucleotide group; Fig. 4a) but did not predict observed mutation frequencies (Fig. 4b; negative binomial generalized linear model, no improvement over null model; *P*~LRT~ = 0.77

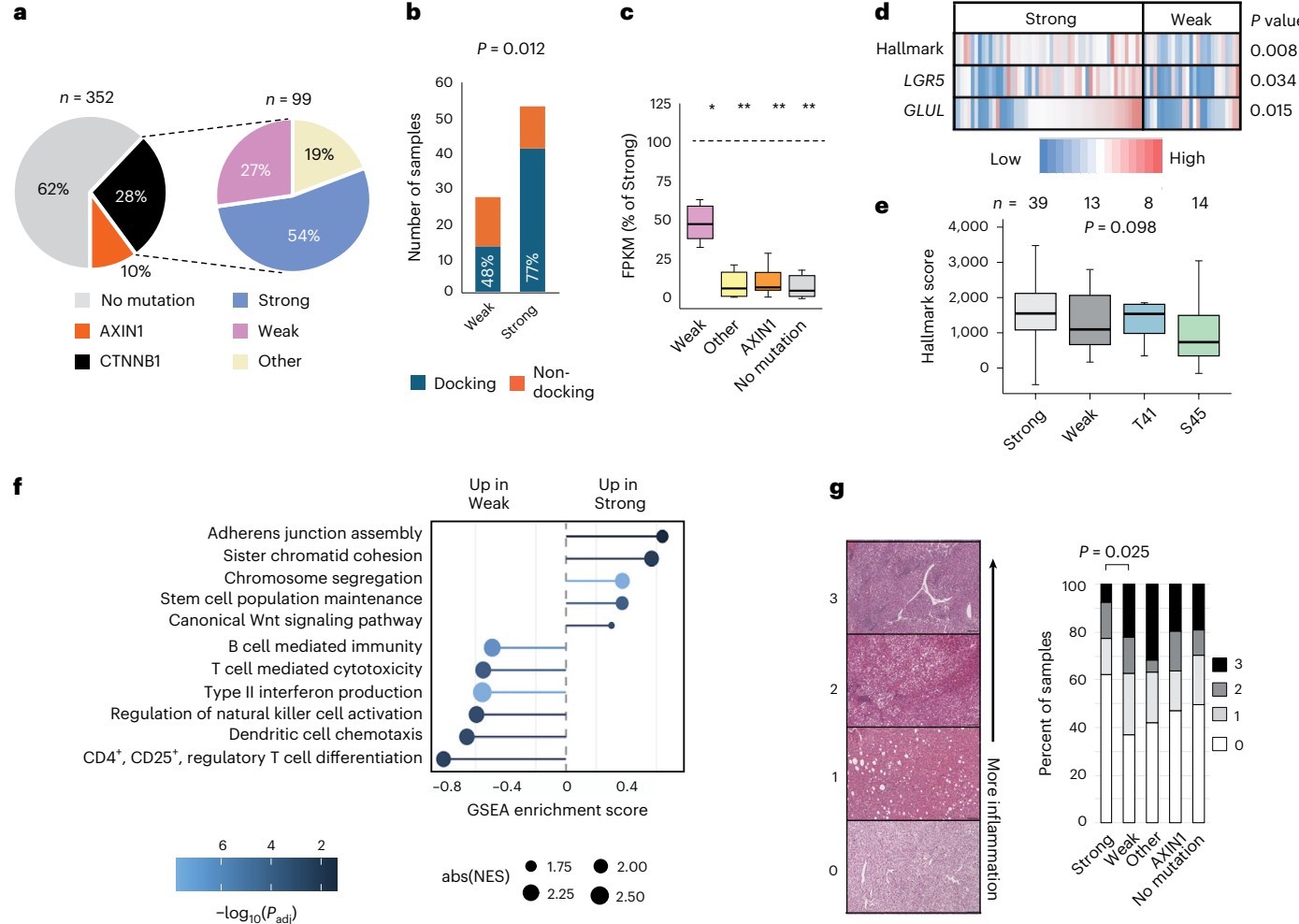

**Fig. 5 | CTNNB1 mutation effect scores predict signaling activation and immune exclusion in HCC. a**, Proportion of HCC samples from TCGA cohort with *CTNNB1* missense or *AXIN1* coding mutations (left). The *CTNNB1* missense mutations are then further divided into three categories (right): strong and weak mutations within the exon 3 hotspot (Supplementary Fig. 6b) and mutations that occurred elsewhere in the gene (other). A total of 18 further samples fell into more than one category or had deletions within *CTNNB1*; therefore, they were excluded. TCGA sample IDs and classifications are listed in Supplementary Table 9. **b**, Stacked histograms show the frequency and proportion of *CTNNB1* exon 3 mutations from TCGA HCC cohort classified as weak or strong that lie in the docking site for β-TRCP (positions 32–37) versus elsewhere in the mutation hotspot. *P* value shows Fisher's exact test (two-sided). **c**, Expression of *n* = 10 β-catenin target genes in HCC stratified by β-catenin pathway mutation status. For each gene, the median expression value was calculated across all samples in the indicated patient group, then expressed as a percentage of the median value for the same gene in the strong group (indicated by the dashed horizontal line at 100%). FPKM, fragments per kilobase per million. Solid horizontal lines show the median percentage value across all ten genes, boxes show the upper and lower quartiles and whiskers show the range. *$P < 4 \times 10^{-3}$, **$P < 1 \times 10^{-4}$ from negative binomial generalized linear model (two-sided), with Tukey's adjustment for a family of five estimates. Gene-level data are shown in Supplementary Fig. 7a. **d**, Heatmap representation of β-catenin pathway activation for TCGA HCC samples with exon 3 hotspot mutations separated into weak and strong categories. *LGR5* and *GLUL* are

individual HCC targets. 'Hallmark' indicates a multi-gene score calculated across 42 genes known to be activated by the accumulation of β-catenin (Hallmark wnt_b-Catenin gene set, MSigDB). *P* values show one-tailed *t*-tests. **e**, Boxplots show Hallmark β-catenin target gene set activation in TCGA tumors with strong or weak mutations in the docking motif, compared to those with mutations at T41 or S45. Horizontal lines show the median value, boxes show the second and third quartiles and whiskers show the range. Samples with copy number gain spanning lower-effect mutations (*n* = 6) were excluded. Differences between groups were not significant in a one-way ANOVA (*P* = 0.098). **f**, Enrichment of Gene Ontology terms in transcripts ranked among the most upregulated in pairwise comparisons between strong and weak *CTNNB1*-mutant HCC samples from TCGA. Terms were selected from the full list shown in Supplementary Table 6. Normalized enrichment scores (NES) show enrichment scores normalized to the size of the gene set. *P* value estimation is based on an adaptive multi-level split Monte Carlo scheme, adjusted for multiple testing using Benjamini–Hochberg correction. GSEA, gene set enrichment analysis. **g**, H&E tumor sections from TCGA were scored using an ordinal scale according to the level of inflammatory infiltrate from zero (no visible immune cells) to three (diffuse or nodular aggregates of immune cells), then scores were compared across patients based on β-catenin pathway mutation status. *P* value shows Fisher's exact test (two-way) for differences between weak and strong groups in the fraction of patients with an immune score of zero versus one or greater. Statistical analysis was not performed on other groups.

and 0.60, respectively). Therefore, tissue-specific *CTNNB1* mutation patterns cannot be explained solely by mutation rates.

We next tested whether tumors arising in different tissues select for hotspot mutations that activate β-catenin signaling to different degrees. Mutations in The Cancer Genome Atlas (TCGA) endometrial

carcinoma cohort (Uterine Corpus Endometrial Carcinoma, UCEC) were enriched for higher MES values (Mann–Whitney test, *P* = $2.2 \times 10^{-5}$), whereas in TCGA HCC cohort (Liver Hepatocellular Carcinoma, LIHC), mutations spanned a broader range (Fig. 4c). This difference persisted following adjustment for tissue-specific background rates of mutation

within coding sequences ($P = 3.2 \times 10^{-5}$). Given that mutation bias does not explain the tissue-specific difference in observed mutation frequencies, we postulate that they arise through natural selection for different optimal β-catenin signaling[2].

Even greater variation in MES values was observed in the larger COSMIC dataset (Supplementary Table 1) across *CTNNB1*-mutant tumors from diverse tissue origins (Fig. 4d; one-way ANOVA, $P < 2 \times 10^{-19}$). Using MES distributions, tissues could be categorized as favoring high-effect mutations (for example, central nervous system), low-effect mutations (for example, kidney) or a broad or bimodal distribution (Fig. 4c). Bimodal distributions, observed in tissues such as large intestine and liver, comprised a lower-effect group, with mutations at S45 and weaker mutations within the β-TRCP docking motif (for example, H36P, D32Y, S33P), and a higher-effect group containing S33, G34, S37 and T41 mutations. These distributions highlight the potential for phenotypic variation arising from different *CTNNB1* mutations among tumors with the same site of origin.

### *CTNNB1* mutation strength correlates with signaling activation and immune exclusion in HCC

To investigate how *CTNNB1* mutation strength influences signaling and immune exclusion in human cancer, we focused on HCC, which shows frequent exon 3 mutations. Patients with single exon 3 missense mutations and available RNA-seq data in TCGA HCC cohort[8] ($n = 80$) were initially stratified into low and high groups based solely on the MES value of their mutation (Supplementary Fig. 6a). However, copy number gains spanning *CTNNB1* enhance signaling, especially for S45 mutant alleles[19], so six patients with copy number gains spanning mutant alleles in the low MES group were reassigned (Supplementary Fig. 6b). This produced 53 patients predicted to have strong and 27 with predicted weak β-catenin pathway activation (Fig. 5a and Supplemental Fig. 6b). For patients in the weak group, 48% had mutations in the docking motif, compared to 77% of the strong group ($P = 0.012$; Fig. 5b)

To assess the effects of other mutations in the β-catenin pathway, we compared transcript levels for ten liver β-catenin target genes in HCC tumors with hotspot missense mutations versus no *CTNNB1* pathway mutation. We included missense mutations in *CTNNB1* outside the hotspot region ('other'; $n = 19$; Fig. 5a), including positions 335, 383 and 387, which disrupt interaction with the destruction complex subunit APC[41], giving a total of 99 *CTNNB1* missense mutation cases in the TCGA HCC cohort ($n = 80$ hotspot, $n = 19$ other). A further 10% of the cohort had coding mutations in *AXIN1* ($n = 36$), another destruction complex subunit and known HCC driver[8,42]. The remainder of the cohort had no *CTNNB1* or *AXIN1* mutation ('no mutation'; $n = 217$).

Tumors in either the weak or strong hotspot class showed markedly higher β-catenin target gene expression compared to the 'no mutation' group (Fig. 5c and Supplementary Fig. 7a). Tumors with 'other' mutations in *CTNNB1*, or *AXIN1* mutations, expressed these targets at levels comparable to the 'no mutation' samples (Fig. 5c and Supplementary Fig. 7a). Therefore, although *AXIN1* and non-hotspot *CTNNB1* mutations contribute to HCC development, their effects on β-catenin target gene expression differ from both weak and strong hotspot mutations[16,42–44].

As predicted by our screen, β-catenin targets were expressed at significantly higher levels in tumors from the strong versus weak patient group (Fig. 5c,d and Supplementary Fig. 7a). This was observed across all ten liver target genes in the curated set (median 49% reduction in transcript levels for the weak group (Fig. 5c and Supplementary Fig. 7a). Tumors with weak exon 3 mutations also displayed significantly lower expression scores for the Hallmark CTNNB1 target gene set from MSigDB (37% median reduction, $P < 0.01$; Fig. 5d). The same trend was confirmed in a second independent HCC cohort (Montironi cohort)[14] (Supplementary Fig. 7b). β-catenin pathway activation in tumors with T41 mutations was comparable to that of strong β-TRCP docking motif variants, with both groups showing non-significant trends

towards higher activation than tumors with the weak docking motif or S45 mutations (one-way ANOVA, $P = 0.098$; Fig. 5e). Overall, weak mutations still activated β-catenin targets, but at intermediate levels compared to strong mutations and non-mutated tumors (Fig. 5c and Supplementary Fig. 7a).

Transcriptome-wide, genes upregulated in strong versus weak HCC samples were significantly enriched for the Gene Ontology term 'canonical Wnt signaling pathway' as well as terms associated with proliferation and stem cell function (Fig. 5f, Supplementary Fig. 8 and Supplementary Tables 5 and 6). Notably, the telomerase subunit *TERT* was among the most significantly upregulated genes (Supplementary Table 5). *TERT* may be a direct transcriptional target of β-catenin regulation[45], and activating mutations in the *TERT* promoter are among the most frequent HCC driver mutations[8,46].

Genes upregulated in tumors with weak versus strong exon 3 mutations were significantly enriched for terms associated with immune cell infiltration (Fig. 5f and Supplementary Table 6). Given that β-catenin signaling activation is a major immune escape pathway in several tumor types, including HCC[15,47], weaker pathway activation may permit greater immune engagement, with potential implications for patient stratification and targeted therapy[15,48,49]. Consistent with this idea, tumors with weak *CTNNB1* mutations showed upregulation of canonical T cell transcripts (Supplementary Fig. 9) to levels similar to tumors lacking *CTNNB1* mutations. In addition, histology confirmed more frequent immune cell infiltration in weak versus strong mutant tumors (65 vs 38% with a score of >1, Fisher's exact test, $P = 0.0245$; Fig. 5g). Altogether, *CTNNB1* exon 3 hotspot MES scores derived from a cell-autonomous reporter assay predict not only β-catenin signaling strength in HCC, but also clinically relevant phenotypes associated with the tumor microenvironment (Fig. 5f,g and Supplementary Fig. 9).

## Discussion

SGE enables genetic variants to be functionally assessed in their native chromosomal context using scalable, multiplexed assays. We developed an SGE screening assay to quantify the impact of all amino acid substitutions in the β-catenin degron on signaling activation, spanning a region that covers >80% of cancer-associated missense mutations. The data explain why particular *CTNNB1* mutations are observed in cancer and others are not, despite disrupting residues known to be critical for degron function.

Our experimental design has three advantages over previous studies that have compared the phenotype of *CTNNB1* mutations[17–19]. First, we tested the function of all 19 alternative amino acids at each position, providing a complete understanding of genotype–phenotype relationships. Second, variants were introduced by genome editing at the endogenous locus, avoiding artifacts associated with ectopic overexpression. Third, the function of each variant was evaluated in parallel, in the same population of primary stem cells with a normal functioning Wnt pathway, providing sensitivity to distinguish subtle phenotypic differences.

Previous studies introduced the concept that distinct *CTNNB1* hotspot mutations activate the canonical Wnt pathway to varying degrees[17–19]. One study[19] analyzed β-catenin target gene activation in benign and malignant liver tumors and proposed that signaling strength depends on mutation position: S45 mutations activate weakly, T41 mutations activate moderately and mutations within the docking site (32–37) activate strongly. Our dataset supports certain aspects of this model, including weaker activity of S45 variants, but also revealed that different substitutions at the same position can have markedly different effects on signaling (Fig. 3a,b). Small side chains at S45 are well tolerated, whereas amino acids at position I35 must support an extended β-sheet-like conformation (Fig. 3f), which is probably required for β-TRCP docking because the same requirement is observed at other β-TRCP-dependent degrons (Fig. 3h and

Supplementary Table 4). Our data thus redefine the consensus motif for β-TRCP docking as DpSGβXpS, where β indicates any residue that supports an extended backbone conformation. At the adjacent position H36, proline is most disruptive (Fig. 2a), probably because of conformational rigidity, explaining why H36P is the only H36 mutation commonly observed in human cancer (Supplementary Table 1).

Our dataset enables functional interpretation for variants of unknown significance within the *CTNNB1* hotspot (Supplementary Table 1) and supports reclassification of variants with known relevance in HCC. We find that common T41 mutations (T41A, T41I) are strong activators, with MES values exceeding the average of docking site mutants (Figs. 3c and 5e and Supplementary Fig. 6). Moreover, about 25% of docking motif mutations in HCC showed relatively low activation, comparable to S45 mutations, which we classify as weak (Fig. 5b and Supplementary Fig. 6). This nearly doubles the proportion of TCGA patients with weak *CTNNB1* mutations.

On average, tumors driven by weak mutations express liver β-catenin target genes at ~50% levels relative to the strong group (Fig. 5c). Importantly, this group is also less likely to be immune-excluded (Fig. 5f,g and Supplementary Fig. 9). We note the similarities between the weak group and the recently described β-catenin 'immune-like' subtype[14]. Weakly activating exon 3 mutations could provide both a novel mechanism and biomarker for this patient group and help to guide strategies for personalized combination-based therapies.

## Study limitations

Our COSMIC database analysis showed that tumors arising in different tissues harbor exon 3 mutations of different predicted strength (Fig. 4d). However, MES values were measured using a reporter gene in mouse ES cells. MES can predict endogenous β-catenin signaling outputs in mouse ES cells (Fig. 2d) and liver (Fig. 5c,d and Supplementary Figs. 4 and 7), and we expect that they will also have predictive power in other cell and tissue contexts. However, further work is required to show this empirically; other β-catenin-dependent tumor types in TCGA were not informative either because of low sample numbers (for example, adrenocortical carcinoma), low frequency of exon 3 mutant tumors (for example, colorectal adenocarcinoma, melanoma) or low diversity of MES values (for example, endometrial carcinoma).

The relationship between exon 3 mutations and signaling outputs in advanced tumors may not reflect earlier stages owing to epistatic effects of co-occurring mutations, together with environmental and metabolic changes that accompany tumor progression. This concern motivated us to conduct the SGE screen in non-transformed primary cells rather than a cancer cell line. We also leave open the possibility that MES values measured in mouse ES cells may lack physiological relevance in certain contexts; for example, where β-catenin is degraded by $SCF^{FBXW11}$ rather than $SCF^{βTRCP}$ (ref. 50).

## Online content

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

¹The Roslin Institute, University of Edinburgh, Edinburgh, UK. ²MRC Human Genetics Unit, Institute of Genetics and Cancer, University of Edinburgh, Edinburgh, UK. ³Department of Human Genetics, Leiden University Medical Centre, Leiden, The Netherlands. ⁴Liver Cancer Translational Research Laboratory, Institut d'Investigacions Biomèdiques August Pi i Sunyer (IDIBAPS), Hospital Clínic, Universitat de Barcelona, Barcelona, Spain. ⁵Facultat de Medicina i Ciències de la Salut, Universitat de Barcelona (UB), Barcelona, Spain. ⁶Transgenic Facility Leiden, Central Animal Facility, Leiden University Medical Centre, Leiden, The Netherlands. ⁷Developmental Biology Program, Sloan Kettering Institute, Memorial Sloan Kettering Cancer Center, New York, NY, USA. ⁸Centre for Inflammation Research, Institute for Regeneration and Repair, University of Edinburgh, Edinburgh, UK. ⁹Cancer Research UK Scotland Institute, Glasgow, UK. ¹⁰Cancer Research Scotland Centre, Edinburgh, UK. ¹¹Cancer Research Scotland Centre, Glasgow, UK. ¹²Mount Sinai Liver Cancer Program, Divisions of Liver Diseases, Tisch Cancer Institute, Icahn School of Medicine at Mount Sinai, New York, NY, USA. ¹³Institució Catalana de Recerca i Estudis Avançats (ICREA), Barcelona, Spain. ¹⁴Koç University Research Center for Translational Medicine (KUTTAM), Istanbul, Turkey. ¹⁵Koç University School of Medicine, Istanbul, Turkey. ✉e-mail: P.Hohenstein@lumc.nl; Andrew.j.wood@ed.ac.uk; Deozdemir@ku.edu.tr

## Methods

This work complied with all relevant ethical regulations. Animal work was approved by Memorial Sloan Kettering Cancer Center's Institutional Animal Care and Use Committee (protocol 03-12-017; principal investigator A.-K. Hadjantonakis). No human data were generated specifically for this study.

### CRISPR design and cloning

All guide RNAs (gRNAs) were designed using the optimized CRISPR design webtool previously hosted by the Feng Zhang laboratory, Benchling (https://benchling.com) and Wellcome Sanger Institute Genome Editing (http://www.sanger.ac.uk/htgt/wge). The gRNAs were cloned into either pSpCas9(BB)-2A-GFP (Addgene plasmid 48138) or pSpCas9(BB)-2A-mCherry[52] and verified by Sanger sequencing.

### Cloning the Puro ΔTK targeting vector

A backbone vector (wild-type β-catenin vector) was generated by amplifying a 5.4 kb region of *Ctnnb1* intron1–6 using primers 5′-GGTTGATACTACCTTGAGTACTC-3′ and 5′-GATTCACAGGGCTGCTA GTG-3′. The amplicon was cloned into the PCR4 TOPO vector using a TOPO TA cloning Kit (Invitrogen). Then, a Gibson cloning reaction was set up, including the wild-type β-catenin vector (amplified using primers 5′-GTGAGGCTTTCTTTGTTGGC-3′ and 5′-GTCAAAAGGC AGAATGAAAACAG-3′), PuroΔTKamplicon (5′-CTGTTTTCATTCTGC CTTTTGACCATAGAGCCCACCGCATCC-3′ and 5′-GCCAACAAAGAAA GCCTCACTACC GGGTAGGGGAGGCG-3′) and Gibson Assembly master mix (NEB) following the manufacturer's guidelines.

### Derivation and maintenance of TCF/Lef-H2B–GFP ES cells

Mouse ES cells were derived from the TCF/Lef-H2B–GFP mouse line[23] using published methods[24]. The cells were then co-transfected with the PuroΔTK targeting vector and *Ctnnb1* sgRNAs 1–4 (5′-TCTGC CTTTTGACGGACATT-3′, 5′-CACCCTCCAGGGCTGCTGTG-3′, 5′-AA AGCCTCACAGGATCCACC-3′ and 5′-TGTTAGAGTTGGGCTAAGGC-3′) using Lipofectamine 2000 (Invitrogen) following the manufacturer's protocol. After Puromycin (1 μg ml⁻¹) selection, individual clones were picked, grown and validated as heterozygously targeted by Sanger sequencing. Cells were then routinely maintained on gelatin-coated flasks in Knockout DMEM (Gibco) supplemented with 10% FBS (GE Healthcare, HyClone), 2 mM L-glutamine (Gibco), 0.1 mM MEM non-essential amino acids, 0.1 mM β-mercaptoethanol (Gibco), 3 μM CHIR99021 (Axon Medchem), 1 μM PD0325901 (Axon Medchem) and leukemia inhibitory factor (hereafter referred to as R2i media).

### Generation of the HDR template library

A double-stranded DNA library was synthesized by Twist Biosciences, with each 200 bp fragment encoding a single distinct amino acid ($n = 20$ per site) spanning codons 31 to 48 of β-catenin. Each fragment was flanked by *BbsI* recognition sites for cloning and included synonymous mutations to enable specific amplification of HDR-edited alleles. Each fragment was then cloned individually into a β-catenin destination vector containing a 5.5 kb β-catenin intron1–6 sequence (5′-GGTTGATACTACCTTGAGTACTC-3′ and 5′-GATTCACAGGGCTGCTAGTG-3′) with two *BbsI* sites flanking the target region. After ligation, the reactions were transformed into Stbl3 competent cells and incubated overnight on a shaker at 37 °C. An equal volume of inoculum from each pool was then combined and used as a starter culture for a single maxiprep plasmid isolation of the pooled HDR template library (Qiagen Maxiprep kit).

### Transfection of HDR template library and flow sorting

TCF/Lef-H2B–GFP reporter cells with heterozygous PuroΔTK knockin at the endogenous *Ctnnb1* locus were maintained in R2i media. A total of 200 × 10⁶ cells were transfected with the pooled HDR template library and PuroΔTK sgRNAs 1–4 (spacer sequences:

5′-TGGGGATGCGGTGGGCTCTA-3′, 5′-CCACCGCATCCCCAGCATGC-3′, 5′-GCCTCCCCTACCCGGTAGTG-3′, 5′-GCCTCACTACCGGGTAGGGG-3′) in 26 six-well plates, using Lipofectamine 2000 (Invitrogen) following the manufacturer's protocol. Small-molecule enhancer L755507 was used at a concentration of 5 μM. On day 3, R2i media was replaced with GMEM (Gibco) supplemented with 10% FBS (GE Healthcare, HyClone), 2 mM L-glutamine (Gibco), 1 mM sodium pyruvate (Gibco), 0.1 mM MEM non-essential amino acids, 0.1 mM β-mercaptoethanol (Gibco) and leukemia inhibitory factor, including FIAU at 0.2 μM to selectively kill cells that had not deleted the TK cassette. On day 5, a single-cell suspension was generated using trypsin, and the cells were sorted based on GFP intensity into six equally logged bins using BD FACS Aria III (BD Biosciences). In a parallel control condition, cells underwent an identical genome editing and selection procedure but were collected without flow sorting (the 'pool' sample). The procedure above was repeated twice, starting with 200 × 10⁶ cells in each case (400 × 10⁶ cells total). Each set of cells went through the same editing, drug selection and flow sorting procedure described above and in Fig. 1d. The number of cells sorted from each bin is listed in Supplementary Table 7.

### DNA isolation and sample processing for deep sequencing

Genomic DNA was isolated using the DNeasy Blood and Tissue kit (Qiagen) according to the manufacturer's protocol. A first round of PCR was performed using a forward primer that annealed upstream of the homology arm (5′-GTGGACATCAGAGGACAACTTG-3′) and a reverse primer that annealed to a region containing the HDR-edited allele (5′-TGTCAACATCTTCTTCTTCGGGA-3′). The entire DNA sample was amplified in several 30-cycle reactions using Q5 Hot Start High-Fidelity 2× Master Mix (NEB), then the amplicons were digested with DpnI (Thermo Scientific), gel-purified and pooled.

For library preparation for the Illumina sequencing platform, a second round of PCR was performed to incorporate Illumina-specific barcode, primer pad, linker and adaptors. The reactions were performed in triplicate for 14 cycles using Q5 Hot Start High-Fidelity 2× Master Mix (NEB), then pooled and purified using AMPure XP beads (Beckman Coulter).

### Targeting and RNA-seq characterization of E14IVtg2a ES cells

The mouse embryonic feeder-free stem cell line E14IVtg2a (E14) was maintained on gelatin-coated flasks in GMEM (Gibco) supplemented with 10% FBS (GE Healthcare, HyClone), 2 mM L-glutamine (Gibco), 1 mM sodium pyruvate (Gibco), 0.1 mM MEM non-essential amino acids, 0.1 mM β-mercaptoethanol (Gibco) and leukemia inhibitory factor. E14 cells were targeted with the puroΔTK vector as described above for the Tcf/Lef-H2B–GFP reporter cells, then the puroΔTK cassette was replaced by HDR donor templates containing individual missense mutations at positions S37, T41 and S45, as shown in Fig. 1d. ES cell colonies were picked and expanded, and clonal lines with heterozygous knockin were identified by Sanger sequencing of PCR amplicons.

Total RNA was isolated from approximately one million cells per sample using the RNAeasy mini kit (Qiagen) according to the manufacturer's instructions. For the generation of cDNA and the RNA-seq library, we adapted the mcSCRB-seq method[53] using previously described modifications[54], starting with 100 ng of total RNA per sample; Illumina paired-end sequencing of the library was then performed on a NovaSeq 6000 (Illumina) following the manufacturer's instructions. After sequencing, Read 1 contained the cDNA information, and read 2 only contained the unique molecular identifier.

A Galaxy workflow consisting of the following tools was used to perform (reads to count) transcriptome analysis. Cutadapt (Galaxy version 4.0+galaxy1; https://usegalaxy.eu/root?tool_id=toolshed.g2.bx. psu.edu/repos/lparsons/cutadapt/cutadapt/4.0+galaxy1) was first used to remove adapter sequences from fastq files. FastQC (Galaxy version 0.73+galaxy0; https://usegalaxy.eu/root?tool_id=toolshed.g2.bx. psu.edu/repos/devteam/fastqc/fastqc/0.73+galaxy0) was then used to

generate the Read Quality reports. UMI-tools extract (Galaxy version 1.1.2+galaxy2; https://usegalaxy.eu/root?tool_id=toolshed.g2.bx.psu.edu/repos/iuc/umi_tools_extract/umi_tools_extract/1.1.2+galaxy2) was used to extract UMI from fastq files. Filter with SortMeRNA (Galaxy version 2.1b.6; https://usegalaxy.eu/root?tool_id=toolshed.g2.bx.psu.edu/repos/rnateam/sortmerna/bg_sortmerna/2.1b.6) was used to filter reads for ribosomal RNAs in metatranscriptomic data. The reads from this filtered fastq files were aligned to mm10 build of mouse genome using RNA STAR the Gapped-read mapper for RNA-seq data (Galaxy version 2.7.8a+galaxy0; https://usegalaxy.eu/root?tool_id=toolshed.g2.bx.psu.edu/repos/iuc/rgrnastar/rna_star/2.7.8a+galaxy0). UMI-tools deduplicate (Galaxy version 1.1.2+galaxy2; https://usegalaxy.eu/root?tool_id=toolshed.g2.bx.psu.edu/repos/iuc/umi_tools_dedup/umi_tools_dedup/1.1.2+galaxy2) was used for deduplication of UMIs. MarkDuplicates (Galaxy version 2.18.2.3; https://usegalaxy.eu/root?tool_id=toolshed.g2.bx.psu.edu/repos/devteam/picard/picard_MarkDuplicates/2.18.2.3) was used to examine aligned records in BAM datasets to locate and filter duplicate molecules. Feature-Counts (Galaxy version 2.0.1+galaxy2; https://usegalaxy.eu/root?tool_id=toolshed.g2.bx.psu.edu/repos/iuc/featurecounts/featurecounts/2.0.1+galaxy2) was used to generate the counts from the alignment data in the BAM files. Bedtools BAM to BED converter (Galaxy version 2.30.0+galaxy1; https://usegalaxy.eu/root?tool_id=toolshed.g2.bx.psu.edu/repos/iuc/bedtools/bedtools_bamtobed/2.30.0+galaxy1) was used to convert the alignment data from BAM to BED file format. Bedtools Genome Coverage (Galaxy version 2.30.0; https://usegalaxy.eu/root?tool_id=toolshed.g2.bx.psu.edu/repos/iuc/bedtools/bedtools_genomecoveragebed/2.30.0) was used to record the genome coverage in bedgraph file format from the BED files with the alignment data. CONVERTER_bedgraph_to_bigwig (Galaxy version 1.0.1; https://usegalaxy.eu/root?tool_id=CONVERTER_bedgraph_to_bigwig) was used to convert the bedgraph file to bigwig file format for upload and visualization of the expression data on the UCSC or IGV genome browser.

The 'mm10 Full' reference genome was used for the entire workflow. The counts table was converted into counts per million for all samples before normalization to z-score values, which were then used to generate a heatmap using the pHeatmap package (v.1.0.12) in RStudio. Unsupervised hierarchical clustering resulted in three main clusters. Differential gene expression analysis was then performed between clusters using the DESeq2 (Galaxy version 2.11.40.8+galaxy0) pipeline, and a final heatmap was generated using the pHeatmap package (v.1.0.12) in RStudio, limited to the list of genes that were differentially expressed between the three clusters.

### Retrieval of COSMIC data for all *CTNNB1*-mutant cancers (V94_38)

Targeted and genome-wide mutations were downloaded from https://cancer.sanger.ac.uk/cosmic/download, filtered by gene *CTNNB1*, from COSMIC release v.94 (28 May 2021). Mutations labeled 'Substitution – Missense' were extracted, excluding one multiple nucleotide polymorphism (p.I35_H36delinsSN).

### SGE read processing and calculation of mutational effect scores

Adaptors were trimmed and paired ends merged with NGmerge (https://github.com/harvardinformatics/NGmerge) to produce single-end reads. Mean sequence quality (Phred) scores were high (>30) across all base positions targeted for mutagenesis, so low-quality reads were not filtered out. Single reads were then aligned with bwa mem (v.0.7.17)[55] to the 162 bp *CTNNB1* reference sequence. A set of reads with a single missense on-target mutation, and no other mutations, was generated. Reads that did not fully cover the region targeted for mutagenesis (58–111 bp) were excluded, as were alignments with insertions and deletions anywhere, no mutations in the target region, mutations only outside the target region, multiple mutations in the target region, synonymous mutations only or mutations resulting in a codon that was not in the repair template library. The remaining reads had precisely one missense mutation specified from the HDR template library.

Read counts were normalized within each of the six experimental bins and two control conditions by dividing the number of reads for each mutation by the total number of filtered reads in that bin, such that the sum of normalized counts for all mutations in the bin was equal to 1. To calculate enrichment values relative to the starting population, the normalized count for each mutation in each bin was then divided by the normalized count for the same mutation in cells that had undergone the same genome editing and selection procedure but had not undergone flow sorting based on GFP. Replicates had high Pearson correlation (0.54–0.89 across GFP bins; Supplementary Fig. 3), so reads were merged for downstream analysis. Plasmid and unselected pool codon frequencies had a Pearson correlation of 0.63. Lastly, each of the six normalized mutant allele frequencies was multiplied by the mean GFP fluorescence value for all cells in the corresponding bin, and then the six values were added together to yield the MES. We noted that a subset of mutant alleles (31 out of 342) was underrepresented in the donor plasmid pool (<0.028% of reads observed versus 0.28% expected), probably as a result of non-uniform growth or processing of bacterial cultures before template pooling. These mutations are marked as 'lower confidence' in Supplementary Table 2. None of the lower confidence mutations were observed among patients in TCGA or Montironi HCC cohorts.

### VEP comparisons

VEP scores were obtained for all β-catenin single amino acid substitutions spanning amino acids 31–48 from 50 different methods, using a published pipeline[26]. The Spearman correlations were then calculated between the outputs of each predictor and the MES values. Note that some predictors only output scores for missense variants possible by single nucleotide changes, meaning that the correlations were calculated from fewer mutations; however, this has been shown to have little effect on overall correlations or relative predictor rankings[56].

### AAIndex

To investigate amino acid properties potentially related to the effects of mutations at I35, we downloaded all 566 indices from the AAIndex database[35,37]. The Spearman correlation was calculated between the values for each amino acid and the MES for each of the 19 substitutions at I35. These correlations are provided in Supplementary Table 8. Peptide sequences for previously reported βTrCP-dependent degrons were extracted from Table S1 of a previous publication[37], then filtered for a precise match to the DSGX motif spanning positions 32–35 of β-catenin (n = 28 peptides, detailed in Supplementary Table 4).

### Stratification of TCGA and Montironi HCC cohorts

Samples from TCGA-LIHC cohort (n = 370) were filtered into one of six groups (Supplementary Table 9): (1) 'strong': those with a hotspot missense mutation with a MES value of >18,000 or MES < 18,000 plus copy number gain defined in cBioportal (n = 53); (2) 'weak': those with a hotspot missense mutation with a MES value of <18,000 and no copy number gain (n = 27); (3) 'other': those with a *CTNNB1* missense mutation outside the hotspot region (n = 19); (4) 'AXIN1': those with any coding mutation in *AXIN1* (n = 36); (5) 'no mutation': those without any coding mutation in *CTNNB1* or *AXIN1* (n = 217); and (6) 'exclude': those with deletions and/or complex mutations in *CTNNB1*, those falling into more than one of groups (1) to (4) and those lacking survival and/or RNA-seq data (n = 18). Group (6) was excluded from further analysis.

For the Montironi cohort, we considered only patients with single missense mutations in the exon 3 hotspot (n = 44). Of those, 31 had mutations classified as strong based on the same classification system described above, including n = 2 with MES < 18,000 plus copy number gain, and 13 had mutations classified as weak.

## Histological scoring of immune cell infiltration

Whole-slide images of haematoxylin and eosin (H&E)-stained, formalin-fixed, paraffin-embedded sections ('diagnostic slide') of TCGA-LIHC cohort were viewed using the NCI GDC Data Portal slide viewer; cases for which only a frozen section image ('tissue slide') was available were not evaluated. Each case was scored by an expert consultant liver histopathologist and National Liver Pathology External Quality Assurance scheme member working at the national liver transplant center (T.K.), blinded to all new experimental data and without accessing any additional TCGA data. Intratumoral inflammation was scored using H&E morphology alone using a four-point ordinal scale[57]: 0, no inflammation; 1, scattered inflammatory cells; 2, focal aggregates of inflammatory cells; and 3, diffuse or nodular aggregates of inflammatory cells.

## RNA-seq and differential expression analysis in HCC cohorts

To compare transcriptomes of HCC samples from the categories shown in Fig. 5b, STAR transcript quantification for TCGA HCC samples ($n = 361$) was downloaded from the GDC Data Portal. Differential expression of samples in each category compared to the weak MES missense mutation was calculated with DeSeq2 (v.1.34)[58] using shrunken log-fold changes. Gene set enrichment analysis was performed to investigate the functional enrichment of the differentially expressed genes identified from RNA-seq data. The ranked list of differentially expressed genes was generated based on the DESeq2 statistics, which takes into account both fold-change and *P* value information. Gene set enrichment analysis was performed on the Gene Ontology database, which consists of three structured, controlled vocabularies. The enrichment analysis was carried out using the ClusterProfiler (v.4.6.2) package in R (v.4.2.0), using the 'gseGO' function with default parameters[59]. The full output of this analysis is shown in Supplementary Table 6. For the analysis of β-catenin target genes, ten known targets (based on published literature) were identified among the 50 most upregulated transcripts in strong versus no mutation tumor groups from TCGA HCC dataset. We modeled expression as predicted by gene identity, HCC subgroup and their interaction using negative binomial models (R package MASS). We then obtained *P* values comparing expression in each HCC subgroup to 'strong' as the baseline using marginal means testing implemented in the R package emmeans.

RNA-seq data from the Montironi cohort were generated and processed as previously described[14]. Hallmark expression scores were generated using the Wnt β-catenin Hallmark signature, using the ssGSEA pipeline implemented in Genepattern[60,61]. For human liver organoid data, normalized RNA-seq counts (fragments per kilobase per million) were downloaded from the Gene Expression Omnibus (Accession ID GSE236490). Expression values were averaged for $n = 2$ biological replicates corresponding to each mutation.

## Deriving nucleotide-level TNC scores from LIHC and UCEC exome data

Exome sequence variants from TCGA-LIHC ($n = 375$) and TCGA-UCEC ($n = 404$) were downloaded from the GDC Data Portal (https://portal.gdc.cancer.gov). Variants called by at least two of the four provided workflows (MuSE, MuTect2, SomaticSniper, VarScan2) were retained. Cases with one or more missense mutations in the 31–48 amino acid target region were extracted (TCGA-LIHC, $n = 82$; TCGA-UCEC, $n = 104$), and the single-nucleotide polymorphisms from these were used to generate tri-nucleotide mutation frequencies with SomaticSignatures[62]. The results shown are based upon data generated by the TCGA Research Network (https://www.cancer.gov/tcga).

## Model to convert tri-nucleotide scores to MLSs

All possible mutational paths from one codon to another, one nucleotide change at a time, were generated for all codon pairs. The MLS of each path was calculated as $\prod_i(\text{freq}_i)$, where $\text{freq}_i$ is the tri-nucleotide context mutational frequency of the $i$th step of the path. The MLS of each codon change in every possible tri-nucleotide context is the sum of all paths between the two codons; similarly, the MLS for each amino acid change in its tri-nucleotide context is the sum of its codon change scores. A worked example is shown in Supplementary Fig. 5b, and MLS scores calculated separately from TCGA-LIHC and TCGA-UCEC are listed in Supplementary Table 11. For statistical testing, we modeled observed mutation frequencies as the result of MLS in the same tumor type, using a negative binomial generalized linear model implemented in the R package MASS. We compared these fits to null models fitting only the mean using likelihood ratio tests.

## Statistics and reproducibility

Statistical tests were performed in R (v.4.2.2) using RStudio (v.2022.12.0). All tests were two-sided unless stated otherwise. Sample size for the SGE screen was determined by the number of possible missense mutations between positions 31 and 48 of β-catenin, and all 342 variants were included. Sample sizes for clinical studies were determined by the number of patient samples available in relevant cohorts that could be unambiguously assigned to one category based on *CTNNB1* pathway mutation status. No blinding or randomization was performed.

## Reporting summary

Further information on research design is available in the Nature Portfolio Reporting Summary linked to this article.

## Data availability

RNA-seq data from genome-edited mouse ES cell lines with individual *CTNNB1* mutations are available from the Gene Expression Omnibus under accession code GSE299075. TCGA data are accessible through cBioportal. RNA-seq and whole-exome sequencing data for the Montironi cohort have been deposited at the European Genome–Phenome Archive using accession code EGAS00001005364. Source data are provided with this paper.

## Code availability

Custom scripts used for data analysis in this manuscript are available at https://doi.org/10.5281/zenodo.17940132 (ref. 63).

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

## Acknowledgements

We thank G. Kudla, I. Adams and L. Boulter for advice and discussions, and thank Y. Ariyurek, S. Kloet and the Leiden Genome Technology Center for support with the RNA-seq work. This work was supported by the Medical Research Council, UK (MR/M010341/1 to P.H., MC_PC_21040 to A.J.W., MC_UU_00035/1 to C.S. and an MRC Unit Award to the MRC Human Genetics Unit); by the BBSRC, UK (BB/P013732/1) to P.H.; by a Wellcome Trust Sir Henry Dale Fellowship (102560/Z/13/Z) to A.J.W.; and by the National Institutes of Health (R01DK12782, R01HD035455, P30CA008748) to A.K.H. J.M.L is supported by grants from European Commission (Horizon Europe-Mission Cancer, THRIVE, Ref. 101136622), the National Institutes of Health (R01-CA273932-01, R01DK56621 and R01DK128289); Samuel Waxman Cancer Research Foundation; the Spanish National Health Institute (MICINN, PID2022-139365OB-I00, funded by MICIU/AEI/10.13039/501100011033 and FEDER); Cancer Research UK (CRUK), Fondazione AIRC per la Ricerca sul Cancro and Fundación Científica de la Asociación Española Contra el Cáncer (FAECC) (Accelerator Award, HUNTER, Ref. C9380/A26813); "la Caixa" Foundation (Agreement LCF/PR/SP23/52950009); Fundación Científica de la Asociación Española Contra el Cáncer (FAECC; Proyectos Generales, Ref. PRYGN223117LLOV; Reto AECC 70% Supervivencia: Ref. RETOS245779LLOV; AECC-IDIBAPS Excellence Program Ref. EPAEC246711CLIN); and the Generalitat de Catalunya/AGAUR (2021 SGR 01347). R.P. was supported by the Fundació de Recerca Clínic Barcelona–IDIBAPS and by a grant from the Spanish National Health Institute (MICINN, PID2022-139365OB-I00). A.M. was supported by Generalitat de Catalunya with an FISDUR fellowship (2021 FISDU 00338) from AGAUR and by mobility grants from the University of Barcelona, Montcelimar Foundation and Acadèmia de Ciències Mèdiques i de la Salut de Catalunya i de Balears Foundation.

## Author contributions

D.D.O., P.H. and A.J.W. jointly supervised the research. A.K., D.D.O., K.S.D., M.K., C.B., J.C., M.M.L. and S.S. performed the wet lab experiments. A.M., K.S.D., M.K., A.M., A.E., G.T., P.G., G.G., H.B., P.H., D.D.O., R.S., A.G.O., R.P., C.S., T.J.K., J.M. and A.J.W. analyzed the data. A.F.V., A.K.H. and J.M.L. contributed reagents, materials and/or analysis tools. A.J.W., P.H., D.D.O., T.G.B., J.M., A.M., J.M.L. and R.P. wrote the paper.

## Competing interests

A.J.W. has received sponsored lecture fees from Altos Labs and serves as a consultant for Gemini Law. T.J.K. has served as a consultant or advisory board member for Resolution Therapeutics, Clinnovate Health, HistoIndex, Fibrofind, Kynos Therapeutics, Perspectum, Concept Life Sciences, Servier Laboratories, Taiho Oncology and Jazz Pharmaceuticals, and has received speakers' fees from Servier Laboratories, Jazz Pharmaceuticals, AstraZeneca, HistoIndex and Incyte Corporation. J.L. has received research support from Genentech & Roche, consulting and sponsored lecture fees from Eisai, Merck, Roche, Genentech, AstraZeneca, Bayer Pharmaceuticals, AbbVie, Sanofi, Moderna, Glycotest, Exelixis and Boehringer Ingelheim, and is on the Data Safety Monitoring Board for Bristol Myers Squibb. All other authors declare no competing interests.

## Additional information

**Correspondence and requests for materials** should be addressed to Peter Hohenstein, Andrew J. Wood or Derya D. Ozdemir.

Derya Ozdemir
Peter Hohenstein

# Reporting Summary

## Statistics

For all statistical analyses, confirm that the following items are present in the figure legend, table legend, main text, or Methods section.

| n/a | Confirmed | |
|---|---|---|
| ☐ | ☒ | The exact sample size (*n*) for each experimental group/condition, given as a discrete number and unit of measurement |
| ☒ | ☐ | A statement on whether measurements were taken from distinct samples or whether the same sample was measured repeatedly |
| ☐ | ☒ | The statistical test(s) used AND whether they are one- or two-sided *Only common tests should be described solely by name; describe more complex techniques in the Methods section.* |
| ☒ | ☐ | A description of all covariates tested |
| ☐ | ☒ | A description of any assumptions or corrections, such as tests of normality and adjustment for multiple comparisons |
| ☐ | ☒ | A full description of the statistical parameters including central tendency (e.g. means) or other basic estimates (e.g. regression coefficient) AND variation (e.g. standard deviation) or associated estimates of uncertainty (e.g. confidence intervals) |
| ☐ | ☒ | For null hypothesis testing, the test statistic (e.g. *F*, *t*, *r*) with confidence intervals, effect sizes, degrees of freedom and *P* value noted *Give P values as exact values whenever suitable.* |
| ☒ | ☐ | For Bayesian analysis, information on the choice of priors and Markov chain Monte Carlo settings |
| ☒ | ☐ | For hierarchical and complex designs, identification of the appropriate level for tests and full reporting of outcomes |
| ☐ | ☒ | Estimates of effect sizes (e.g. Cohen's *d*, Pearson's *r*), indicating how they were calculated |

*Our web collection on statistics for biologists contains articles on many of the points above.*

## Software and code

Policy information about availability of computer code

| | |
|---|---|
| Data collection | No software was used. |
| Data analysis | Scripts used to convert Illumina amplicon sequencing data to counts for each missense mutation in each bin for the saturation mutagenesis screen, and to calculate mutational likelihood scores from exome sequencing data, are available at https://git.ecdf.ed.ac.uk/igmmbioinformatics/betacatenin-saturation-screen<br><br>A Galaxy workflow consisting of the following tools were used to perform (reads to count) transcriptome analysis of edited E14IVtg2a cells. toolshed.g2.bx.psu.edu/repos/devteam/fastqc/fastqc/0.73+galaxy0 toolshed.g2.bx.psu.edu/repos/iuc/umi_tools_extract/umi_tools_extract/1.1.2+galaxy2 toolshed.g2.bx.psu.edu/repos/rnateam/sortmerna/bg_sortmerna/2.1b.6 toolshed.g2.bx.psu.edu/repos/iuc/rgrnastar/rna_star/2.7.8a+galaxy0 toolshed.g2.bx.psu.edu/repos/iuc/umi_tools_dedup/umi_tools_dedup/1.1.2+galaxy2 toolshed.g2.bx.psu.edu/repos/devteam/picard/picard_MarkDuplicates/2.18.2.3 toolshed.g2.bx.psu.edu/repos/iuc/bedtools/bedtools_bamtobed/2.30.0+galaxy1 toolshed.g2.bx.psu.edu/repos/iuc/featurecounts/featurecounts/2.0.1+galaxy2 toolshed.g2.bx.psu.edu/repos/iuc/bedtools/bedtools_genomecoveragebed/2.30.0 CONVERTER_bedgraph_to_bigwig.<br><br>CTNNB1 mutations were annotated from WES data from the Montironi cohort as described in Montironi et al., Gut, 2023, and each amino acid change was annotated and classified as STRONG or WEAK based on the scoring system described in the present study. RNA-Seq data from the Montironi cohort was analysed via single-sample Gene Set Enrichment Analysis (ssGSEA) pipeline implemented in GenePattern, available at: https://www.genepattern.org/modules/docs/ssGSEAProjection/4/. |

For manuscripts utilizing custom algorithms or software that are central to the research but not yet described in published literature, software must be made available to editors and reviewers. We strongly encourage code deposition in a community repository (e.g. GitHub). See the Nature Portfolio guidelines for submitting code & software for further information.

## Data

Policy information about availability of data

All manuscripts must include a data availability statement. This statement should provide the following information, where applicable:
- Accession codes, unique identifiers, or web links for publicly available datasets
- A description of any restrictions on data availability
- For clinical datasets or third party data, please ensure that the statement adheres to our policy

RNA-Seq data from genome edited mESC lines with individual CTNNB1 mutations are available from the Gene Expression Omnibus (GEO) using accession code GSE299075. TCGA data are accessible via cBioportal. The RNAseq and Whole-exome sequencing data for the Montironi cohort have been deposited at the European Genome-Phenome Archive (EGA) using accession code EGAS00001005364. RNA Seq data from base edited human fetal liver organoids are available from GEO using accession ID GSE236490.

## Research involving human participants, their data, or biological material

Policy information about studies with human participants or human data. See also policy information about sex, gender (identity/presentation), and sexual orientation and race, ethnicity and racism.

| | |
|---|---|
| Reporting on sex and gender | A large majority of hepatocellular carcinoma patients with CTNNB1 exon 3 mutations are male, which provided limited power to detect sex differences in mutation strength. We did not detect significant sex differences in the proportion of exon 3 hotspot mutations classed as high versus low effect in males versus females in the TCGA cohort. Information on the sex of human fetal liver material used by Geurts et al (PMID: 37591832) to generate liver organoids used in this study was not available. |
| Reporting on race, ethnicity, or other socially relevant groupings | Race and ethnicity of hepatocellular carcinoma cohorts used in this study is described in Wheeler et al 2017 (PMID: 28622513) and Montironi et al 2023 (PMID: 35197323) Race and ethnicity were not tested as variables in this study. |
| Population characteristics | N/A |
| Recruitment | No patients were recruited specifically for this study. |
| Ethics oversight | This study used only public human data. No ethical oversight was in place. |

Note that full information on the approval of the study protocol must also be provided in the manuscript.

# Field-specific reporting

Please select the one below that is the best fit for your research. If you are not sure, read the appropriate sections before making your selection.

☒ Life sciences ☐ Behavioural & social sciences ☐ Ecological, evolutionary & environmental sciences

For a reference copy of the document with all sections, see nature.com/documents/nr-reporting-summary-flat.pdf

# Life sciences study design

All studies must disclose on these points even when the disclosure is negative.

| | |
|---|---|
| Sample size | Sample size for the mutational scanning experiment was determined by the number of possible amino acid substitutions spanning CTNNB1 amino acid positions 31 - 48. Sample size for HCC cohorts was determined by the number of patients available with relevant mutations and associated transcriptome data. Sample size for the human organoid study was determined by the number of samples available in GEO dataset GSE236490. |
| Data exclusions | Clones that showed prolonged difficulty to recover following thawing, and were therefore likely subject to selection for specific cell populations or additional culture-induced changes, were excluded from downstream analysis. |
| Replication | Survival analysis based on the TCGA cohort was included in our original submission and biorxiv preprint. This failed to replicate in the Montironi cohort so was excluded from the final version. All other attempts at replication were successful. |
| Randomization | Randomisation was not performed |
| Blinding | Blinding was not performed. |

# Reporting for specific materials, systems and methods

We require information from authors about some types of materials, experimental systems and methods used in many studies. Here, indicate whether each material, system or method listed is relevant to your study. If you are not sure if a list item applies to your research, read the appropriate section before selecting a response.

## Materials & experimental systems

| n/a | Involved in the study |
|-----|----------------------|
| ☒ | ☐ Antibodies |
| ☐ | ☒ Eukaryotic cell lines |
| ☒ | ☐ Palaeontology and archaeology |
| ☒ | ☐ Animals and other organisms |
| ☒ | ☐ Clinical data |
| ☒ | ☐ Dual use research of concern |
| ☒ | ☐ Plants |

## Methods

| n/a | Involved in the study |
|-----|----------------------|
| ☒ | ☐ ChIP-seq |
| ☒ | ☐ Flow cytometry |
| ☒ | ☐ MRI-based neuroimaging |

## Antibodies

| Antibodies used | *Describe all antibodies used in the study; as applicable, provide supplier name, catalog number, clone name, and lot number.* |
|---|---|
| Validation | *Describe the validation of each primary antibody for the species and application, noting any validation statements on the manufacturer's website, relevant citations, antibody profiles in online databases, or data provided in the manuscript.* |

## Eukaryotic cell lines

Policy information about cell lines and Sex and Gender in Research

| Cell line source(s) | Primary mouse embryonic stem cells expressing the TCF:GFP reporter were derived at Memorial Sloan Kettering Cancer Centre. E14IVtg2a cell lines originated from the Cell line cryobank at the MRC Human Genetics Unit, Edinburgh. The origin of human fetal liver organoids is described in Geurts et al 2023 (PMID: 37591832). |
|---|---|
| Authentication | TCF:GFP mESC lines were authenticated based on the response of the fluorescent reporter to CHIR99021. No authentication was performed for mouse E14IVtg2a cells. |
| Mycoplasma contamination | All embryonic stem (ES) cell work was conducted in dedicated culture rooms operating under a strict mouse pathogen monitoring policy. Routine health surveillance was performed using the IDEXX IMPACT II mouse pathogen panel, which screens for 18 murine pathogens, including Mycoplasma spp. and Mycoplasma pulmonis. Cultures were confirmed negative for mycoplasma contamination prior to experimentation. |
| Commonly misidentified lines (See ICLAC register) | N/A |

## Palaeontology and Archaeology

| Specimen provenance | *Provide provenance information for specimens and describe permits that were obtained for the work (including the name of the issuing authority, the date of issue, and any identifying information). Permits should encompass collection and, where applicable, export.* |
|---|---|
| Specimen deposition | *Indicate where the specimens have been deposited to permit free access by other researchers.* |
| Dating methods | *If new dates are provided, describe how they were obtained (e.g. collection, storage, sample pretreatment and measurement), where they were obtained (i.e. lab name), the calibration program and the protocol for quality assurance OR state that no new dates are provided.* |

☐ Tick this box to confirm that the raw and calibrated dates are available in the paper or in Supplementary Information.

| Ethics oversight | *Identify the organization(s) that approved or provided guidance on the study protocol, OR state that no ethical approval or guidance was required and explain why not.* |
|---|---|

Note that full information on the approval of the study protocol must also be provided in the manuscript.

## Animals and other research organisms

Policy information about studies involving animals; ARRIVE guidelines recommended for reporting animal research, and Sex and Gender in Research

| Laboratory animals | *For laboratory animals, report species, strain and age OR state that the study did not involve laboratory animals.* |
|---|---|

| | |
|---|---|
| Wild animals | *Provide details on animals observed in or captured in the field; report species and age where possible. Describe how animals were caught and transported and what happened to captive animals after the study (if killed, explain why and describe method; if released, say where and when) OR state that the study did not involve wild animals.* |
| Reporting on sex | *Indicate if findings apply to only one sex; describe whether sex was considered in study design, methods used for assigning sex. Provide data disaggregated for sex where this information has been collected in the source data as appropriate; provide overall numbers in this Reporting Summary. Please state if this information has not been collected. Report sex-based analyses where performed, justify reasons for lack of sex-based analysis.* |
| Field-collected samples | *For laboratory work with field-collected samples, describe all relevant parameters such as housing, maintenance, temperature, photoperiod and end-of-experiment protocol OR state that the study did not involve samples collected from the field.* |
| Ethics oversight | *Identify the organization(s) that approved or provided guidance on the study protocol, OR state that no ethical approval or guidance was required and explain why not.* |

Note that full information on the approval of the study protocol must also be provided in the manuscript.

# Clinical data

Policy information about clinical studies

All manuscripts should comply with the ICMJE guidelines for publication of clinical research and a completed CONSORT checklist must be included with all submissions.

| | |
|---|---|
| Clinical trial registration | *Provide the trial registration number from ClinicalTrials.gov or an equivalent agency.* |
| Study protocol | *Note where the full trial protocol can be accessed OR if not available, explain why.* |
| Data collection | *Describe the settings and locales of data collection, noting the time periods of recruitment and data collection.* |
| Outcomes | *Describe how you pre-defined primary and secondary outcome measures and how you assessed these measures.* |

# Dual use research of concern

Policy information about dual use research of concern

## Hazards

Could the accidental, deliberate or reckless misuse of agents or technologies generated in the work, or the application of information presented in the manuscript, pose a threat to:

| No | Yes | |
|----|-----|---|
| ☒ | ☐ | Public health |
| ☒ | ☐ | National security |
| ☒ | ☐ | Crops and/or livestock |
| ☒ | ☐ | Ecosystems |
| ☒ | ☐ | Any other significant area |

## Experiments of concern

Does the work involve any of these experiments of concern:

| No | Yes | |
|----|-----|---|
| ☒ | ☐ | Demonstrate how to render a vaccine ineffective |
| ☒ | ☐ | Confer resistance to therapeutically useful antibiotics or antiviral agents |
| ☒ | ☐ | Enhance the virulence of a pathogen or render a nonpathogen virulent |
| ☒ | ☐ | Increase transmissibility of a pathogen |
| ☒ | ☐ | Alter the host range of a pathogen |
| ☒ | ☐ | Enable evasion of diagnostic/detection modalities |
| ☒ | ☐ | Enable the weaponization of a biological agent or toxin |
| ☒ | ☐ | Any other potentially harmful combination of experiments and agents |

# Plants

**Seed stocks**
*Report on the source of all seed stocks or other plant material used. If applicable, state the seed stock centre and catalogue number. If plant specimens were collected from the field, describe the collection location, date and sampling procedures.*

**Novel plant genotypes**
*Describe the methods by which all novel plant genotypes were produced. This includes those generated by transgenic approaches, gene editing, chemical/radiation-based mutagenesis and hybridization. For transgenic lines, describe the transformation method, the number of independent lines analyzed and the generation upon which experiments were performed. For gene-edited lines, describe the editor used, the endogenous sequence targeted for editing, the targeting guide RNA sequence (if applicable) and how the editor was applied.*

**Authentication**
*Describe any authentication procedures for each seed stock used or novel genotype generated. Describe any experiments used to assess the effect of a mutation and, where applicable, how potential secondary effects (e.g. second site T-DNA insertions, mosiacism, off-target gene editing) were examined.*

# ChIP-seq

## Data deposition

☐ Confirm that both raw and final processed data have been deposited in a public database such as GEO.

☐ Confirm that you have deposited or provided access to graph files (e.g. BED files) for the called peaks.

**Data access links**
*May remain private before publication.*
*For "Initial submission" or "Revised version" documents, provide reviewer access links. For your "Final submission" document, provide a link to the deposited data.*

**Files in database submission**
*Provide a list of all files available in the database submission.*

**Genome browser session**
(e.g. UCSC)
*Provide a link to an anonymized genome browser session for "Initial submission" and "Revised version" documents only, to enable peer review. Write "no longer applicable" for "Final submission" documents.*

## Methodology

**Replicates**
*Describe the experimental replicates, specifying number, type and replicate agreement.*

**Sequencing depth**
*Describe the sequencing depth for each experiment, providing the total number of reads, uniquely mapped reads, length of reads and whether they were paired- or single-end.*

**Antibodies**
*Describe the antibodies used for the ChIP-seq experiments; as applicable, provide supplier name, catalog number, clone name, and lot number.*

**Peak calling parameters**
*Specify the command line program and parameters used for read mapping and peak calling, including the ChIP, control and index files used.*

**Data quality**
*Describe the methods used to ensure data quality in full detail, including how many peaks are at FDR 5% and above 5-fold enrichment.*

**Software**
*Describe the software used to collect and analyze the ChIP-seq data. For custom code that has been deposited into a community repository, provide accession details.*

# Flow Cytometry

## Plots

Confirm that:

☐ The axis labels state the marker and fluorochrome used (e.g. CD4-FITC).

☐ The axis scales are clearly visible. Include numbers along axes only for bottom left plot of group (a 'group' is an analysis of identical markers).

☐ All plots are contour plots with outliers or pseudocolor plots.

☐ A numerical value for number of cells or percentage (with statistics) is provided.

## Methodology

**Sample preparation**
*Describe the sample preparation, detailing the biological source of the cells and any tissue processing steps used.*

**Instrument**
*Identify the instrument used for data collection, specifying make and model number.*

**Software**
*Describe the software used to collect and analyze the flow cytometry data. For custom code that has been deposited into a community repository, provide accession details.*

| Cell population abundance | *Describe the abundance of the relevant cell populations within post-sort fractions, providing details on the purity of the samples and how it was determined.* |
| Gating strategy | *Describe the gating strategy used for all relevant experiments, specifying the preliminary FSC/SSC gates of the starting cell population, indicating where boundaries between "positive" and "negative" staining cell populations are defined.* |

☐ Tick this box to confirm that a figure exemplifying the gating strategy is provided in the Supplementary Information.

# Magnetic resonance imaging

## Experimental design

| Design type | *Indicate task or resting state; event-related or block design.* |
| Design specifications | *Specify the number of blocks, trials or experimental units per session and/or subject, and specify the length of each trial or block (if trials are blocked) and interval between trials.* |
| Behavioral performance measures | *State number and/or type of variables recorded (e.g. correct button press, response time) and what statistics were used to establish that the subjects were performing the task as expected (e.g. mean, range, and/or standard deviation across subjects).* |

## Acquisition

| Imaging type(s) | *Specify: functional, structural, diffusion, perfusion.* |
| Field strength | *Specify in Tesla* |
| Sequence & imaging parameters | *Specify the pulse sequence type (gradient echo, spin echo, etc.), imaging type (EPI, spiral, etc.), field of view, matrix size, slice thickness, orientation and TE/TR/flip angle.* |
| Area of acquisition | *State whether a whole brain scan was used OR define the area of acquisition, describing how the region was determined.* |

Diffusion MRI     ☐ Used     ☐ Not used

## Preprocessing

| Preprocessing software | *Provide detail on software version and revision number and on specific parameters (model/functions, brain extraction, segmentation, smoothing kernel size, etc.).* |
| Normalization | *If data were normalized/standardized, describe the approach(es): specify linear or non-linear and define image types used for transformation OR indicate that data were not normalized and explain rationale for lack of normalization.* |
| Normalization template | *Describe the template used for normalization/transformation, specifying subject space or group standardized space (e.g. original Talairach, MNI305, ICBM152) OR indicate that the data were not normalized.* |
| Noise and artifact removal | *Describe your procedure(s) for artifact and structured noise removal, specifying motion parameters, tissue signals and physiological signals (heart rate, respiration).* |
| Volume censoring | *Define your software and/or method and criteria for volume censoring, and state the extent of such censoring.* |

## Statistical modeling & inference

| Model type and settings | *Specify type (mass univariate, multivariate, RSA, predictive, etc.) and describe essential details of the model at the first and second levels (e.g. fixed, random or mixed effects; drift or auto-correlation).* |
| Effect(s) tested | *Define precise effect in terms of the task or stimulus conditions instead of psychological concepts and indicate whether ANOVA or factorial designs were used.* |

Specify type of analysis:     ☐ Whole brain     ☐ ROI-based     ☐ Both

| Statistic type for inference (See Eklund et al. 2016) | *Specify voxel-wise or cluster-wise and report all relevant parameters for cluster-wise methods.* |
| Correction | *Describe the type of correction and how it is obtained for multiple comparisons (e.g. FWE, FDR, permutation or Monte Carlo).* |

## Models & analysis

n/a | Involved in the study
--- | ---
☐ ☐ | Functional and/or effective connectivity
☐ ☐ | Graph analysis
☐ ☐ | Multivariate modeling or predictive analysis

**Functional and/or effective connectivity**

*Report the measures of dependence used and the model details (e.g. Pearson correlation, partial correlation, mutual information).*

**Graph analysis**

*Report the dependent variable and connectivity measure, specifying weighted graph or binarized graph, subject- or group-level, and the global and/or node summaries used (e.g. clustering coefficient, efficiency, etc.).*

**Multivariate modeling and predictive analysis**

*Specify independent variables, features extraction and dimension reduction, model, training and evaluation metrics.*

