## [Peer Review File · Nature Genetics]

Mutational scanning reveals oncogenic CTNNB1 mutations have diverse effects on signalling

Corresponding Author: Dr Andrew Wood

Version 0:

Decision Letter:

26th Sep 2023

Dear Andrew,

Your Technical Report, "Mutational scanning reveals oncogenic CTNNB1 mutations have diverse effects on signalling and clinical traits" has now been seen by 3 referees. You will see from their comments copied below that while they find your work of considerable potential interest, they have raised quite substantial concerns that must be addressed. In light of these comments, we cannot accept the manuscript for publication, but would be very interested in considering a revised version that addresses these serious concerns.

In brief, while the reviews overall appreciate the application of DMS to the important oncogene beta-catenin, they do not sound convinced that the overall novelty - as yet - justifies publication in Nature Genetics.

Reviewer #1 is the most unsupportive, saying "the study in many ways validates what is known but does not extend knowledge or mechanistic insight in a significant manner." They make a number of suggestions for improvement, notably performing the screen in various cancer cell line contexts, as well as important technical issues.

Referee #2 is more supportive, but also asks for further experimental validation - in agreement with Referee #1.

Reviewer #3 also sounds supportive; their requests are primarily for further methodological detail, and regarding the comparison to EVE.

In our reading of these reviews, we have placed extra weight on Reviewer #1, as they are the biological system expert, and we think that their suggestion for further screens in other cell line models is an excellent one. We note that this aligns with Referee #2's request for further experimental validation, and therefore we suggest that this be made the top priority for a revision. The remainder of the requests are, of course, also important - especially those concerning technical aspects of the screen - but without the improved biological novelty that Reviewer #1 is seeking, we think a revision will have difficulty in review.

We hope you will find the referees' comments useful as you decide how to proceed. If you wish to submit a substantially revised manuscript, please bear in mind that we will be reluctant to approach the referees again in the absence of major revisions.

To guide the scope of the revisions, the editors discuss the referee reports in detail within the team, including with the chief editor, with a view to identifying key priorities that should be addressed in revision and sometimes overruling referee requests that are deemed beyond the scope of the current study. We hope that you will find the prioritised set of referee points to be useful when revising your study. Please do not hesitate to get in touch if you would like to discuss these issues further.

If you choose to revise your manuscript taking into account all reviewer and editor comments, please highlight all changes in the manuscript text file. At this stage we will need you to upload a copy of the manuscript in MS Word .docx or similar editable format.

*2) If you have not done so already please begin to revise your manuscript so that it conforms to our Technical Report format instructions, available here. Refer also to any guidelines provided in this letter.

*3) Include a revised version of any required Reporting Summary: <https://www.nature.com/documents/nr-reporting-summary.pdf>

Please be aware of our guidelines on digital image standards.

Link Redacted

If you wish to submit a suitably revised manuscript we would hope to receive it within 6 months. If you cannot send it within this time, please let us know. We will be happy to consider your revision so long as nothing similar has been accepted for publication at Nature Genetics or published elsewhere. Should your manuscript be substantially delayed without notifying us in advance and your article is eventually published, the received date would be that of the revised, not the original, version.

Nature Genetics is committed to improving transparency in authorship. As part of our efforts in this direction, we are now requesting that all authors identified as 'corresponding author' on published papers create and link their Open Researcher and Contributor Identifier (ORCID) with their account on the Manuscript Tracking System (MTS), prior to acceptance. ORCID helps the scientific community achieve unambiguous attribution of all scholarly contributions. You can create and link your ORCID from the home page of the MTS by clicking on 'Modify my Springer Nature account'. For more information please visit please visit www.springernature.com/orcid.

Thank you for the opportunity to review your work.

Sincerely,

Michael Fletcher, PhD
Senior Editor, Nature Genetics

ORCID: 0000-0003-1589-7087

Referee expertise:

Referee #1: Wnt signalling, cancer.

Referees #2, #3: deep mutational scans, VEP.

Reviewers' Comments:

Reviewer #1:

Remarks to the Author:

In this study, Krishnan and colleagues perform a comprehensive mutational scan of exon 3 CTNNB1 mutations using a Tcf reporter system. A mutational effect score is devised to score the activity (GFP levels from reporter) of each mutant allele and link genotype to phenotypic effect. In silico data analyses and mining from COSMIC and TCGA data-sets are then used to bin CTNNB1 mutation-scores as weak/strong and correlate with survival/molecular features in different cancer types.

Overall the paper is clearly written and utilises an innovative screening approach to efficiently scan through hundreds of plausible CTNNB1 mutations in the exon 3 hotspot. Given that most CTNNB1 mutations and their outputs are already well described in the literature, the study in many ways validates what is known but does not extend knowledge or mechanistic

insight in a significant manner.

Specific comments:

The screen is performed in the context of murine ESCs – given that the screen is mono-allelic, can the authors comment on allelic expression of beta-catenin and whether this may influence the screening result.

Since a major point of the study is that CTNNB1 mutations may correlate with oncogenic function in a tissue specific manner, it would be interesting to perform the screens in a variety of tissue specific lines.

The oncogenic functions of CTNNB1 mutations are solely inferred from GFP levels in an artificial reporter system. Validation studies using endogenous beta-catenin targets would be useful. Since beta-catenin activity can be regulated at different levels, measuring GFP alone is insufficient for drawing broad conclusions.

Figure 1 presents an in silico analysis/review of COSMIC data. This data is easily accessed/visualised directly from COSMIC. The spectrum of CTNNB1 mutations in cancers has been thoroughly described and reviewed in the literature. Would be more appropriate to present this as a Supplementary Figure and reference the literature.

CTNNB1 mutations have already been linked to favourable prognosis in HCC. The literature should be cited in this regard (doi: 10.3892/mco.2015.569, doi: 10.1016/S0002-9440(10)64590-7).

Reviewer #2:

Remarks to the Author:

A. Krishnan and colleagues present a clever saturation editing system for the functional characterization of CTNNB1 cancer variants and careful correlative analyses using TCGA data.

B. While others (Shendure etc) have presented saturation editing approaches in the past, this group appears to have developed a clever innovation on the approach that enables assaying a non-essential gene. This was achieved by first integrating a negative selection marker into the gene, and then performing editing. The authors could emphasize the significance of this strategy, as it enables screening any gene using saturation genome editing, whereas in the past only essential genes in HAP1 cells could be assayed. In addition, this paper focuses more on the biology of CTNNB1 variants rather than the method per se.

C. Overall, the screening methodology was sound and performed in a robust manner.

D. Appropriate statistical tests were used throughout, although I have some question about the MLS/MES analysis (see below).

E. Conclusions are valid and interesting, although I would have expected at least some minimal experimental validation of the screen results (see below).

F. Suggested improvements:

1) Figure 2/S2 - I suggest moving the full schematic from S2 to figure 2.

2) Figure 3A - stats should be shown to support the assertion on p. 7 line 183-184 that mutations at position 45 tend to activate reporter gene expression to a lower level than sites 32/33/34/37

3) Figure 4 - Authors show that different tumor types harbor different background mutation rates, but that these do not fully explain the spectrum of mutations observed in each tumor type. Then, in Fig 4C, authors show that mutations within each tumor type vary with respect to their mutation function score (MES). However, this result may be a result of the aforementioned different mutational spectrum. I think the authors need to perform a multivariate analysis or otherwise control for the differing mutational spectrum within different tissue lineages (i.e. the plot in Figs 4C-D should be corrected with respect to differing MLS).

4) The authors should clarify and justify why they focus specifically on hepatocellular carcinoma in Figure 5, rather than analyzing other tumor types. What is observed in other cancer types where CTNNB1 mutations are prevalent?

5) Figure callouts on p11-12 appear incorrect and should be rechecked. For example the callout on line 332 and the callout on line 340

6) Regarding experimental validation, it would enhance the rigor of the study to also show that individually cloned variants (or single cell clones of the screened population) confirm the results of the sequencing-based screen. Authors could validate variants from within each bin of CTNNB1 activity.

7) The paper presents data from multiple tumor types, then focuses on HCC. However it is not clear if the scores from ES cells best represent the function of CTNNB1 in one tissue type or another. The interpretation given for strong vs. weak mutations in a given tumor type is that HCC may select for weakly activating CTNNB1 mutations. However, it seems possible that in HCC cells the phenotypes of these mutations might be different (for example if a CTNNB1 interacting protein is expressed in HCC but not ES or vice versa). The conclusions of the paper would be enhanced by a counter screen in HCC cells or other evidence that the CTNNB1 mutations result in the same functional groups in HCC cells.

G. References are appropriate.

H. The manuscript writing was clear and appropriate.

Reviewer #3:

Remarks to the Author:

Krishnan et al. describe a small multiplex assay of variant effect to identify mutations in the β -catenin degron that disrupt protein-protein interactions and lead to stabilization of β -catenin. The authors use a modified saturation genome editing approach to introduce mutations into an engineered mouse ESC line. Then they stratify mutations in the degron by transcriptional activation using a standard flow-seq strategy with a β -catenin-responsive reporter. They identify mutations in codon positions known to be important for E3 ligase binding and phosphorylation suggesting the assay is behaving as designed. The authors compare their data to an unsupervised variant effect predictor and note that EVE does not do as well as the assay at identifying hotspot codons (as one might expect). The authors find that many mutations found in cancer are bimodally distributed between weak and strong activation of β -catenin and that there was a difference in survival between these two populations. The authors then characterize the phenotypic differences between these two populations and note differential gene expression and immune cell invasion. In summary, this is a neat use of mutational scanning to explore a long-standing question in β -catenin-driven cancer. I do have points that should be addressed before publication.

Please address the following:

The methods section is missing key information.

- How many clones?
- How many cells transfected?
- What was the replicate structure? Multiple transfections or one pool of cell and replicate flow sorts?
- How many replicate experiments were performed and over what time frame?
- Scatter plots of replicates are required. Can you throw out the worst behaved replicates and improve data quality?
- How many cells were sorted in each experiment? How many in each bin?
- How much DNA (converted to genome equivalents) was used to amplify in the PCR?
- The Pearson R range for replicate experiments in the methods section does not match the range reported in the results section.
- What parameters were used for NGmerge (were low quality reads filtered?)
- Why were the synonymous variants filtered out? They can serve as a nice neutral control.

"Replacement frequency" is confusing to me and not a common way of referring to mutation abundance.

Scatter plot for S3 please.

The framing of the comparison to EVE is off base. Of course, the assay is going to be better at identifying GOF mutations, you've set up an assay to select for them. The fact that EVE can identify what are essentially GOF mutations at all is impressive (Livesey and Marsh 2023). This part of the paper should be edited to reflect that and not that the assay won a straw-man competition vs. a computational variant effect predictor.

Also, the second paragraph that discusses EVE's erroneous identification of two codon positions (G38 and P44) that aren't selected for in the assay. EVE uses multiple sequence alignments to query evolutionary conservation of those codons. The assay in this manuscript assesses a single function of β -catenin. It is entirely possible that evolution has constrained these amino acids for other reasons not assessed in this assay. Your assessment that the assay can better stratify β -catenin mutation important for cancer may be correct, but likely not because EVE is wrong.

Line 240. The word unusual should be removed. The whole point of the functional data y'all worked so hard to generate was to do this analysis, correct?

The TCGA and COSMIC analyses are outside my expertise but look compelling to me.

Lea Starita
University of Washington

Version 1:

Decision Letter:

Our ref: NG-TR63256R

7th Oct 2025

Dear Dr. Wood,

Thank you for submitting your revised manuscript "Mutational scanning reveals oncogenic CTNNB1 mutations have diverse effects on signalling" (NG-TR63256R). It has now been seen by the original referees and their comments are below. The reviewers find that the paper has improved in revision, and therefore we'll be happy in principle to publish it in Nature Genetics, pending minor revisions to satisfy the referees' final requests and to comply with our editorial and formatting guidelines.

Sincerely,

Margot Brandt, PhD
Senior Editor
Nature Genetics
<https://orcid.org/0000-0002-9434-794X>

Reviewer #1 (Remarks to the Author):

The authors have addressed most of the reviewer comments through additional experiments and supporting data sets. I am still rather concerned about the paucity of direct data provided in human model systems. For example, in mouse - loss of APC leads to tumours in the small intestine, not colon highlighting genetic difference in this pathway between mouse and human with consequently divergent phenotype. To this point, the relevance to human cancer is still somewhat lacking as the screens are performed in mouse centric models and human evidence is mainly correlative and inferential from human cancer or external organoid gene expression datasets. While the authors make a technical case for performing the screens in primary lines versus cancer lines, there is a missed opportunity as I believe the screens could have been performed or at least more direct functional evidence provided in human models such as organoid or primary lines.

Reviewer #2 (Remarks to the Author):

The authors have done an impressive job responding to reviewer comments, in particular the mutation likelihood score correction/analysis and the validation in clonal ESC lines. Overall, all of my major concerns have been addressed.

Minor comments:

- In Figure 4C it should be stated in the legend if the wilcoxon test shown was or was not adjusted for mutation likelihood.
- In Figure 5E and G, authors should put the statistical test mentioned in the text in the figure and legend as well.

Reviewer #3 (Remarks to the Author):

I am satisfied with the revisions.

Response to Reviewers

We would like to thank the three reviewers for their insightful comments on our manuscript. Their suggestions for additional experiments and clarifications have helped us to improve the work significantly. Below, we provide a point-by-point response addressing their concerns. To briefly summarise, we have:

- validated our reporter-based scoring system on endogenous β -catenin target genes in a series of clonal genome edited mESC lines and human fetal liver organoids.
- improved our system for patient stratification in hepatocellular carcinoma and analysed a second patient cohort. Both of these changes further support the relevance of our mESC assay to β -catenin signalling outputs in human disease.
- Reorganised Figure 1, added substantial new data to Figures 2 and 5, added two new Supplemental Figures (S4, S6), and added methodological details requested by reviewer 3.
- Re-written the introduction, results and discussion sections to make it clear how our findings extend current knowledge, and where further work is needed in the future.

We hope that the three reviewers appreciate the substantial effort that has gone into these revisions.

Reviewer #1:

Remarks to the Author:

In this study, Krishnan and colleagues perform a comprehensive mutational scan of exon 3 CTNNB1 mutations using a Tcf reporter system. A mutational effect score is devised to score the activity (GFP levels from reporter) of each mutant allele and link genotype to phenotypic effect. In silico data analyses and mining from COSMIC and TCGA data-sets are then used to bin CTNNB1 mutation-scores as weak/strong and correlate with survival/molecular features in different cancer types.

Overall the paper is clearly written and utilises an innovative screening approach to efficiently scan through hundreds of plausible CTNNB1 mutations in the exon 3 hotspot. Given that most CTNNB1 mutations and their outputs are already well described in the literature, the study in many ways validates what is known but does not extend knowledge or mechanistic insight in a significant manner.

We thank the reviewer for recognising the innovative nature of our screening approach. We agree that the subset of CTNNB1 mutations observed in tumours have been well described by genome sequencing studies, and that previous studies have introduced the concept that different missense mutations can elicit different effects on signalling (Austin et al., 2008; Provost et al., 2003; Rebouissou et al., 2016). However, only a very small number of mutations have been directly compared using cellular assays. Ours is the first study to systematically compare all missense variants within the CTNNB1 mutation hotspot. Below, we explain how our findings **refine** the model proposed by the most relevant previous study (Rebouissou et al., 2016) while also

generating **new mechanistic insights**. Because our manuscript was submitted to
Nature Genetics as a Technical Report, we emphasise how these insights were enabled
by a unique study design which has inherent value from a technical standpoint, as
noted by other reviewers.

A prior study from the Zucman-Rossi group (Rebouissou et al., 2016) measured β -
catenin target gene activation in benign and malignant liver tumour samples with
different *CTNNB1* mutations. A small subset of exon 3 hotspot mutations ($n = 3$) were
also tested in HCC cell lines using a transient cDNA overexpression system. This study
concluded that missense mutations in the β -TRCP docking motif (positions 32 – 37)
exert the strongest effects on β -catenin signalling, followed by mutations at T41, then
S45. In other words, **current knowledge is based on the position in the β -catenin**
**polypeptide at which the mutation occurs**.

In our study, the phenotype of all 342 missense substitutions was compared in
an isogenic, non-transformed primary stem cell system with a normal functioning wnt
pathway. We edited the endogenous chromosomal locus, avoiding technical artifacts
associated with ectopic overexpression or biological variability associated with inter-
tumour heterogeneity. Compared to overexpression of cDNAs in cancer cell lines, our
study design in many ways better represents the context in which mutations would be
expressed during early tumour evolution. This allowed us to develop
**genotype/phenotype correlations based on both amino acid position, and on the**
**identity of the new amino acid**. To our knowledge, our manuscript is the first to do this
for any amino acid position in β -catenin, and we do it systematically across positions
covering 81% of all *CTNNB1* missense substitutions in the TCGA Hepatocellular
Carcinoma dataset, and 88% of *CTNNB1* missense mutations across all tumour types
in the COSMIC database.

Reassuringly, our study confirmed some of the conclusions from (Rebouissou et al.,
2016). Specifically, our findings align for position S45, where cancer-associated
mutations cause relatively low signalling outputs in both mESCs (Figure 3A, 3B, 3C in
our revised submission) and in HCC (Figure 5E). We also agree that mutations
associated with stronger signalling outputs are enriched in the docking motif. However,
our findings call for refinement of the current model in two critical ways:

- 1) Based on our mutational scanning assay, we show that **mutations in the docking**
**motif are not all strong activators**: several found commonly in cancer (e.g, D32V,
D32A, D32Y, H36P) activate signalling to levels comparable to S45 mutations in our
stem cell assay (Reviewer Figure 1, Figure S6 in our revised manuscript). Tumour
transcriptome data are consistent with this finding (Reviewer Figure 2, Figure 5E).
We therefore propose that this subset of docking site mutations, which includes
approximately 25% of all docking site mutations in the TCGA HCC cohort, should be
reclassified as WEAK activators (Reviewer Figure 3).
- 2) We show that T41 mutations observed in TCGA (T41A, T41I) activate to a level that is
higher than average for TCGA mutations in the docking motif (Reviewer Figure 1,
Figure S6). Rather than assigning T41 mutations into their own Intermediate class
(Rebouissou et al., 2016), **we propose that T41A and T41I should be reclassified**
**as STRONG activators**. Other rare, but nonetheless recurrent, T41 mutations

(T41N, T41S – see Table S1 in our revised submission) should be reclassified as
 WEAK activators.

Reviewer Figure 1 (also Figure S6)

Lollipop plot shows the distribution and frequency of MES values for CTNNB1 exon 3 mutations observed in the TCGA HCC cohort. Colours indicate the position of each mutation based on the categories proposed in (Rebouissou et al., 2016), illustrating that docking motif mutations have diverse effects on signalling, and that T41 mutations activate relatively strongly.

Reviewer Figure 2 (also Figure 5E)

Box and whisker plot shows the level of CTNNB1 pathway activation in RNA-Seq data from TCGA HCC tumours with mutations in the docking motif classed as STRONG or WEAK, compared to any mutation at position T41 or S45. Samples with copy number gain spanning any WEAK effect mutant allele (n = 6, see response to Reviewer 1 point 1 below) were excluded from this analysis. Boxes denote the 25th and 75th percentile values where horizontal black lines denote the median. Hallmark score is calculated from a multi-gene signature of 42 transcripts known to be upregulated by increased CTNNB1 abundance (MSigDB ID M5895). Differences between groups are not significant at $p < 0.05$ based on a One-Way ANOVA test, but trends are consistent with the mESC assay data shown in Reviewer Figure 1

Our revised classification system almost doubled the number of TCGA tumours in the WEAK category (n = 14 patients with S45 only, n = 27 patients including S45 and WEAK docking mutants). This is significant given that this expanded patient group, which our manuscript systematically defines for the first time, does not show the immune exclusion phenotype associated with STRONG exon 3 mutations (Figure 5F, 5G and Figure S9 of our revised manuscript).

Reviewer Figure 3 (also Figure 5A)

Stacked histograms summarise the revised classification of CTNNB1 exon 3 mutations in TCGA. The addition of specific docking site mutations to the WEAK group almost doubles the number of samples in this category. The addition of T41 mutations to the STRONG group leads to a ~20% increase. P-value indicates Fisher’s exact test showing enrichment of STRONG mutations within versus outside the docking motif.

We also point out that the novelty in our work is not restricted to the HCC-centric analysis outlined above, but also significantly extends mechanistic knowledge on structure/function relationships within the β -catenin degron which are likely relevant to β -catenin regulation in other tumour types and during normal development.

For example, we show that:

- alternative amino acids are well tolerated at serine 45 provided they have a small side chain. This has mechanistic implications for the CK1 α ‘priming’ model of phosphodegron function, because well-tolerated amino acids such as glycine and cysteine are not targeted by kinases of the serine/threonine family (Johnson et al., 2023).
- Amino acids at position 35 must support an extended β -sheet-like conformation in the local secondary structure (Figure 3F). This is likely required during β -TRCP docking, because we found that the same secondary structure requirement is observed at other β -TRCP-dependent degrons elsewhere in the proteome (Figure 3G & 3H). Thus, we propose that the consensus motif for the recognition of substrates by β -TRCP should be redefined from DpSGXXpS (or DpSG ϕ XpS, where ϕ indicates any amino acid with a hydrophobic side chain and pS denotes phospho-serine) to DpSG β XpS, where β indicates any residue that supports an extended backbone conformation.
- Proline is the most disruptive residue at position 36 (H36P), likely due to its conformational rigidity, explaining why no other substitution is commonly observed at this site in human cancer.

We hope that this helps to clarify the ways in which our manuscript extends current knowledge and mechanistic insight. To ensure that these points are clear to the reader in the revised text, we have rewritten sections of the Results (lines 222 – 295, lines 344 – 396) and discussion (lines 423 - 479), and added the three Figure panels shown in Reviewer Figures 1, 2 and 3 at positions indicated in their respective legends.

Specific comments:

1) The screen is performed in the context of murine ESCs – given that the screen is mono-allelic, can the authors comment on allelic expression of beta-catenin and

163 whether this may influence the screening result.

We thank the reviewer for this comment. The saturation genome editing screen was
'monoallelic' in that we measured variant effects on one allele in the presence of a
second wildtype allele that remained unedited. It was deliberately and necessarily
designed this way to avoid introducing more than one mutation per cell, which would
prevent us from linking genotype and cellular phenotype.

Nonetheless, we considered the possibility that the presence of a *wild-type* allele in our
monoallelic screen might influence signalling readouts by partially buffering the effect
of weaker activating variants. While this is a valid concern, our screen was designed to
mimic the heterozygous state seen in many *CTNNB1*-mutant tumours. Moreover, we
observed robust separation of variant effects, including between WEAK and STRONG
activators, and between variants which are, versus are not, commonly observed in
human cancer. We are therefore confident that the dynamic range of our assay
captures functional differences despite the presence of a *wildtype* allele.

2) Since a major point of the study is that *CTNNB1* mutations may correlate with
oncogenic function in a tissue specific manner, it would be interesting to perform the
screens in a variety of tissue specific lines.

We thank the reviewer for this suggestion and agree that it would be interesting to
repeat the saturation mutagenesis screen in other physiologically relevant cell contexts.
However, as outlined further below, there are technical constraints that make this task
extremely challenging, and biological constraints that would complicate data
interpretation even if it were technically feasible.

Below, we:

1) Explain why the mESC system used in our original submission was uniquely
suitable for the screen for both biological and technical reasons.

2) Outline our efforts during the revision period to repeat the screen in primary
neural progenitor cells and murine embryonic fibroblasts, and why this proved
unsuccessful.

3) Describe new data and analyses added to our revised manuscript, including
transcriptome analysis from two hepatocellular carcinoma patient cohorts, and
a series of human fetal liver organoid lines, that support the relevance of mESC
data to other cell contexts.

4) Describe text added to the discussion to clarify that MES values predict levels of
β -catenin signalling mESCs, and in normal and malignant human liver, but that
further experimental work is needed to determine whether their relevance is
universal.

**Biological relevance of mESCs:** *CTNNB1* mutations arise early in tumourigenesis, in
non-transformed cells with an intact canonical Wnt signalling pathway. In contrast,
tumour cell lines derived from β -catenin-dependent cancers will, by definition, have
abnormal β -catenin regulation at baseline, even if *CTNNB1* itself is wild type. This limits
the dynamic range of reporter gene activation, particularly for weaker effect mutations,

and hence the sensitivity of our assay to distinguish variant effects. Conversely, cell
lines derived from non-Wnt dependent tumours lack physiological relevance. We
selected primary embryonic stem cells for our screen because they retain normal,
endogenous β -catenin pathway function. Unmodified by mutation or differentiation,
pluripotent mESCs provide a ‘blank slate’ against which abnormal phenotypes can be
measured. While tissue stem cell populations might better model the impact of
individual tumour types, mESC-based scores are tissue-agnostic.

For this reason, other groups have previously used mESCs to quantify the impact of *Apc*
mutations on β -catenin activity levels using a similar reporter system to ours (Kielman
et al., 2002). *Apc* mutations of differing strength defined using this system were
subsequently shown to determine tissue-specific tumourigenesis in mice (Bakker et al.,
2013).

**Technical Feasibility of mESCs for multiplex homology-directed repair:** The screen
requires diploid cells that express the reporter and respond homogeneously to a β -
catenin-activating stimulus (e.g. CHIR99021, or any individual exon 3 mutation). The
cells need to be efficiently transfectable and have high rates of homology directed DNA
repair following Cas9-mediated double strand break formation: properties that are
found in mouse embryonic stem cells but rarely in other primary cells or cancer cell
lines. It took substantial time and effort to engineer the ‘multiplex ready’ primary cell
line and develop the screen, as outlined in Figure 1D of our revised manuscript.

**Establishing additional multiplex-ready primary cell lines**

In an attempt to measure the consequences of *Ctnnb1* mutations in additional mouse
primary cell types, we injected multiplex-ready mESCs into blastocysts before transfer
into pseudopregnant females to generate chimeric embryos (Reviewer Figure 4). Neural
progenitor cells (NPCs) and Murine Embryonic Fibroblasts (MEFs) were isolated from
dissociated embryonic day 14.5 chimeras as described (Pollard, 2013), and pure
populations of ‘multiplex-ready’ cells were isolated by selection on puromycin using
concentrations that killed 100% of wildtype MEF and NPC cultures. However, whereas
mESCs expressed the TCF:H2B:GFP reporter homogeneously in response to the GSK3 β
inhibitor CHIR99021 at 3 μ M, neither neural stem cells nor MEFs activated the β -catenin
reporter at similar concentration and responded in a highly heterogeneous manner
when exposed to higher concentrations, with only a minority showing any response
(Reviewer Figure 4). Efforts to introduce individual *Ctnnb1* exon 3 mutations via
homology-directed repair and FIAU selection were unsuccessful. We therefore
conclude that NPCs and MEFs are not suitable cell types to repeat the screen for both
biological (stochastic response to β -catenin induction) and technical (challenging to
edit at high efficiency) reasons.

**New transcriptomic data analysis supports relevance of MES beyond mESCs:**

Despite the challenges highlighted above, we agree that it is necessary to show that
MES values are biologically relevant beyond the context of mESCs. In our original
submission, this was achieved via integration with clinical HCC data; specifically RNA
Seq transcriptomics from TCGA. In our revised submission, we have improved our

classification system, added data from a second, independent, HCC cohort, and from
 base-edited human fetal liver organoids.

**Reviewer Figure 4:** *Neural Progenitor Cells (NPCs) and Murine Embryonic Fibroblasts (MEFs)*
 *respond heterogeneously to CTNNB1 induction. Top: Derivation of NPCs and MEFs via*
 *blastocyst injection of multiplex-ready mouse embryonic stem cells. Bottom: Flow cytometry*
 *measurement of Tcf:Lef:H2B:GFP reporter expression in response to CHIR99021 treatment.*
 *Note that mESCs were routinely cultured in the presence of CHIR99021 and long-term*
 *withdrawal was incompatible with viability. NSCs were routinely cultured in EGF and FGF*
 *before CHIR99021 treatment for 48 hours. Longer treatments with CHIR99021, either in the*
 *presence or absence of EGF and FGF, were incompatible with viability.*

**Improved classification:** (Rebouissou et al., 2016) showed that a subset of HCC
 samples with S45 mutations (WEAK activators) can show copy number gain, leading to
 increased β -catenin signalling, which could impact the predictive power of MES values
 in cancer. Our original submission failed to take this into account.

To investigate this, we screened the TCGA LIHC cohort to identify *CTNNB1*
 mutant tumours classed as WEAK in our original analysis (including S45 and WEAK
 docking motif lesions), to look for evidence of copy number gain. We found 6 / 33 where
 this was true, including 4 mutations at S45 and a further 2 at D32. These 6 samples
 indeed showed higher activation of β -catenin targets, on average, relative to the
 remaining 27 WEAK activating samples (Reviewer Figure 5).

We therefore reclassified these 6 samples into the STRONG group of exon 3
 mutant tumours and found, consistent with previous work (Rebouissou et al., 2016),
 that this substantially improved the predictive value of MES on β -catenin pathway
 activation in clinical HCC samples (Reviewer Figure 6). This highlights the importance of
 *CTNNB1* gene dosage in the prediction of signalling outputs, supporting the use of
 genome editing versus overexpression systems for the accurate comparison of
 *CTNNB1* variant function.

Hallmark wnt_b_catenin

Reviewer Figure 5: Heatmap showing the distribution of β -catenin target gene activation, based on the Hallmark wnt_b_catenin multi-gene signature, among $n = 33$ HCC tumours in TCGA with exon 3 mutations classed as WEAK in our original submission. Black arrows indicate samples for which the mutant CTNNB1 locus is subject to copy number gain. Blue = low expression, Red = high expression.

Reviewer Figure 6: Boxplot compares the median transcript levels for the $n = 10$ liver β -catenin targets shown in Figure S7A of our revised manuscript. For each gene, median normalised transcript count values for samples in the WEAK group are shown as red points, normalised to the median value of samples in the STRONG group for the same gene. The “WEAK all” group includes the $n = 6$ samples with CN gain spanning the mutant CTNNB1 allele. In the “WEAK CN gain removed” group, these 6 samples were reassigned to the STRONG group.

New HCC cohort: In our revised submission, we have also analysed two additional RNA-Seq datasets, including a second, independent HCC cohort (Montironi et al., 2023) which shows a similarly strong association between MES and β -catenin target gene activation (Reviewer Figure 7, Figure 5D, Figure S7B).

Reviewer Figure 7 (also Figure 5D and Figure S7B): Heatmap representation of β -catenin pathway activation for HCC patients with exon 3 hotspot mutations from TCGA (left) and Montironi (right) cohorts, separated into WEAK and STRONG categories. LGR5 and GLUL are individual β -catenin target genes in HCC. ‘Hallmark’ indicates a multi-gene score calculated across 42 genes known to be upregulated by the accumulation of beta-catenin (Hallmark_wnt_b-Catenin gene set, MSigDB). p-values indicate 1-tailed t-tests.

**Fetal organoid clonal lines:** In addition, we have analysed published RNA-Seq data
 generated by Hans Clevers' laboratory (Geurts et al., 2023) focusing on β -catenin target
 gene expression levels in human fetal liver organoids that were subjected to base
 editing to generate 4 different exon 3 mutations, expressed in a homozygous manner
 from the endogenous locus. Positive correlations are observed between MES values
 and transcript levels for 9/10 liver target genes (Reviewer Figure 8, Figure S4).

 **Reviewer Figure 8 (also Figure S4):** β -catenin target gene expression in base edited human fetal liver
 organoids **A.** Schematic outline of the procedure followed by (Geurts et al 2023) to introduce
 homozygous exon 3 mutations using cytidine or adenine base editors. **B.** Scatter plots show correlation
 between mutation effect scores for the 4 mutations shown in panel A and normalised transcript counts of
 liver β -catenin target genes. Normalised transcript counts (FKPM) were plotted as the mean of $n = 2$
 biological replicate RNA-Seq experiments.

 Thus, although it was not feasible to repeat the saturation genome editing screen in a
 variety of cell types, our revised manuscript nonetheless contains several new lines of
 evidence that MES values predict endogenous β -catenin signalling beyond the context
 of mESCs, specifically in liver, which is of particular clinical relevance and also the
 main focus of our study. We believe it is reasonable to assume that the scores will have
 predictive value in at least some, if not most, other tissue contexts.

Limitations of the current study

We also acknowledge that caution should be exercised with extrapolation of MES
 values to tissues where their relevance is yet to be formally tested. We have added two
 paragraphs at the end of our manuscript to spell out the limitations of our study,
 including mechanisms through which MES values might not be predictive in all
 biological contexts. The text is provided below:

Study limitations

Our analysis of data from the COSMIC database showed that tumours arising in different
 tissue sites harbour exon 3 mutations of different predicted strength (Figure 4D). However, it
 is important to note that MES values were measured using a reporter gene in mouse ESCs.
 Based on RNA-Seq analyses of β -catenin target genes in cell lines, organoids and tumours

*with individual exon 3 mutations, it is clear that MES can predict endogenous β -catenin*
*signalling outputs in mESCs (Figure 2D) and liver (Figure S4, Figure 5C, 5D, Figure S7),*
*and we expect that they will have predictive power in at least some other cell and tissue*
*contexts relevant to early tumorigenesis. However, further work is required to show this*
*empirically; other β -catenin -dependent tumour types in TCGA were not informative either*
*due to low sample number (e.g. adrenocortical carcinoma), low frequency of exon 3 mutant*
*tumours (e.g. colorectal adenocarcinoma, melanoma) or low diversity of MES values (e.g.*
*endometrial carcinoma).*

*We also note that the relationship between exon 3 mutations and signalling outputs in*
*advanced tumours may not reflect earlier stages due to epistatic effects of co-occurring*
*mutations, together with environmental and metabolic changes that accompany tumour*
*progression. Indeed, this concern motivated us to conduct the saturation genome editing*
*screen in non-transformed primary cells rather than a cancer cell line. We also leave open*
*the possibility that MES values measured in mESCs may lack physiological relevance in*
*certain contexts, for example, where β -catenin is degraded via SCF^{FBXW11} rather than*
*$SCF^{\beta TRCP}$ (Wong et al., 2025).*

**3) The oncogenic functions of CTNNB1 mutations are solely inferred from GFP levels in**
**an artificial reporter system. Validation studies using endogenous beta-catenin targets**
**would be useful. Since beta-catenin activity can be regulated at different levels,**
**measuring GFP alone is insufficient for drawing broad conclusions.**

We agree and thank the reviewer for pointing this out. To address the comment, we
performed genome editing in E14 mESCs to generate a series of 15 clonal cell lines with
heterozygous knock in of 11 different exon 3 mutations, including several WEAK and
STRONG lesions, focused on sites with the highest mutation frequencies in human
cancer (Reviewer Figure 9A, Figure 2B). To measure signalling outputs, we performed
RNA Sequencing on these lines, and included unedited parental cells treated with
increasing doses of the GSK3b inhibitor CHIR99021 as a positive control.

These experiments show the following:

- 1) In unbiased hierarchical clustering based on whole transcriptomes, cell lines
engineered with mutations of similar MES range cluster together (Reviewer
Figure 9B & 9C, Figure 2B & 2C in the revised manuscript).
- 2) Transcript levels for β -catenin target genes, identified in mESCs based on their
dose response to CHIR99021 treatment (Reviewer Figure 9C / Figure 2C), show
positive correlations with MES values across the 15 RNA Seq datasets (Reviewer
Figure 9D / Figure 2D).

Reviewer Figure 9 (also Figure 2): Mutation effect scores predict endogenous target gene expression **A**. Schematic to illustrate derivation of clonal mESC lines genome edited to express individual *Ctnnb1* exon 3 mutations. **B** Unsupervised clustering of *Ctnnb1* mutant clones based on global RNA-Seq expression profiling shows co-clustering of clones with mutations of similar MES values (purple bar). **C & D**. Ten endogenous β -catenin target genes were selected based on the high correlation (>0.85) between transcript expression and dose of the GSK3 β inhibitor CHIR99021 (**E**). Transcript expression also correlated with MES values of the relevant exon 3 mutations (**D**). R-values are from Pearson correlations, asterisks and red text indicate $p < 0.05$.

These points, together with our analysis of endogenous β -catenin targets in two
independent HCC cohorts (Reviewer Figure 7, also Figure 5D and Figure S7B of our
revised submission) and 4 base edited human fetal liver organoid lines (Reviewer Figure
8 / Figure S4), show that scores derived from the TCF:GFP:H2B reporter are translatable
to endogenous gene targets not only in mESCs, but also in normal and malignant
hepatocytes.

4) Figure 1 presents an in silico analysis/review of COSMIC data. This data is easily accessed/visualised directly from COSMIC. The spectrum of CTNNB1 mutations in cancers has been thoroughly described and reviewed in the literature. Would be more appropriate to present this as a Supplementary Figure and reference the literature.

We acknowledge this point and have restructured Figure 1 such that it no longer
 includes only in silico analysis of COSMIC data, but now also describes the saturation
 genome editing screen and shows new data. We feel that the tissue-specific mutation
 profile from COSMIC is an important point for the reader to understand the rationale for
 the screen, so we prefer to keep one panel in the main figure to illustrate this (Figure 1C
 in our resubmission).

5) CTNNB1 mutations have already been linked to favourable prognosis in HCC. The
 literature should be cited in this regard (doi: 10.3892/mco.2015.569, doi:
 10.1016/S0002-9440(10)64590-7).

We thank the reviewer for raising these papers, which report that patients with any
 *CTNNB1* mutation survived for longer, on average, than patients who lacked *CTNNB1*
 mutation. During the revision period, we gained access to a second HCC cohort with
 available outcome data (Montironi et al., 2023), and have reviewed this, together with
 data from TCGA, in consultation with experts in the field of clinical HCC research (Prof.
 Josep Llovet group, who are included as coauthors on our revised submission). In
 contrast to the studies cited by the reviewer, in neither of the two cohorts did we
 observe more favourable prognosis in HCC patients with, versus without, *CTNNB1*
 mutation (Reviewer Figure 10).

**Reviewer Figure 10:** Relationship between *CTNNB1* mutation status and survival in HCC
 Kaplan Meier analysis showing differences in overall survival between patients from the TCGA (left)
 and Montironi (right) cohorts with any mutation in *CTNNB1* (*CTNNB1* mutant) versus no mutation
 in *CTNNB1* (*CTNNB1* WT). *p* values are from Log rank tests.

We also expanded the comparisons from our original submission between patients
 with *CTNNB1* exon 3 mutations that were classified as WEAK versus STRONG,
 examining several different survival endpoints (Overall Survival, Recurrence Free
 Survival, Progression Free Survival, Time to Progression, representative plots are shown
 in Reviewer Figure 11). Although individual endpoints appear to show significant
 associations in individual cohorts, we found no consistent or reproducible associations
 between mutation strength and clinical outcome across the two cohorts. In addition, the small
 number of events preclude the multivariate analysis that would be required to provide robust
 insights on the independent prognostic value of *CTNNB1* mutation strength.

Importantly, the absence of a strong correlation between mutation strength and
 clinical outcome does not undermine the validity or utility of our classification system. Our
 STRONG/WEAK categorization is based on functional effects on β -catenin signalling: a
 mechanistically relevant and well characterized pathway in HCC. The fact that this molecular
 classification does not translate directly into prognostic differences likely reflects the
 multifactorial nature of HCC progression and the complexity of survival as a clinical endpoint,
 which is influenced by diverse biological, etiological, and treatment-related variables. Given
 these complexities, and to maintain scientific rigor and clarity of focus, we have decided not
 to include survival analysis in our revised submission.

 **Reviewer Figure 11: Relationship between CTNNB1 exon 3 mutation class and survival in**
 **HCC** Kaplan Meier analysis compares outcome measures between patients with STRONG and
 WEAK missense mutations in exon 3 in the TCGA (left) and Montironi (right) HCC cohorts. P
 values indicate log-rank tests. Overall survival data were available for both cohorts, but other
 outcome measures are not directly comparable between cohorts due to differences in study
 design.

**Reviewer 2**

466 A. Krishnan and colleagues present a clever saturation editing system for the functional
characterization of CTNNB1 cancer variants and careful correlative analyses using
TCGA data.

B. While others (Shendure etc) have presented saturation editing approaches in the
past, this group appears to have developed a clever innovation on the approach that
enables assaying a non-essential gene. This was achieved by first integrating a negative
selection marker into the gene, and then performing editing. The authors could
emphasize the significance of this strategy, as it enables screening any gene using
saturation genome editing, whereas in the past only essential genes in HAP1 cells could
be assayed. In addition, this paper focuses more on the biology of CTNNB1 variants
rather than the method per se.

C. Overall, the screening methodology was sound and performed in a robust manner.

D. Appropriate statistical tests were used throughout, although I have some question
about the MLS/MES analysis (see below).

E. Conclusions are valid and interesting, although I would have expected at least some
minimal experimental validation of the screen results (see below).

We thank the reviewer for their positive comments and recognition of the technical
innovation in our screening approach.

F. Suggested improvements:

1) Figure 2/S2 - I suggest moving the full schematic from S2 to figure 2.

We agree and have restructured the first two figures in response to this comment and
reviewer 1 comment 4. A schematic depicting the full screening procedure is now
included as Figure 1D.

2) Figure 3A - stats should be shown to support the assertion on p. 7 line 183-184 that
mutations at position 45 tend to activate reporter gene expression to a lower level than
sites 32/33/34/37

We performed One Way ANOVA tests with post hoc Tukey HSD to look for differences in
MES values between substitutions at 32/33/34/37 versus 41 versus 45, for both the
single nucleotide substitution group (n = ~6 missense substitutions per codon) and all
mutations (n = 19 per codon). In both cases, significant differences were observed for
the 32/33/34/37 versus 45 comparison (p < 0.001 for all mutations, p < 0.05 for the
single mutation group), but not for 41 versus 32/33/34/37 or for 41 versus 45. The
results of this test are reported in the legend for Figure 3 panel B of our revised
submission.

3) Figure 4 - Authors show that different tumor types harbor different background
mutation rates, but that these do not fully explain the spectrum of mutations observed
in each tumor type. Then, in Fig 4C, authors show that mutations within each tumor
type vary with respect to their mutation function score (MES). However, this result may
be a result of the aforementioned different mutational spectrum. I think the authors

need to perform a multivariate analysis or otherwise control for the differing mutational
spectrum within different tissue lineages (i.e. the plot in Figs 4C-D should be corrected
with respect to differing MLS).

In line with the reviewer's suggestion, we repeated our analysis of differences in
mutational effect scores between hepatocellular carcinoma and endometrial
carcinoma from TCGA while correcting for tumour-type-specific mutational likelihoods.
This had minimal effect on the conclusion (Mann Whitney $p = 2.2 \times 10^{-5}$ without
correction versus $p = 3.2 \times 10^{-5}$ with correction). COSMIC tumour samples shown in
Figure 4D lack the genome sequencing data necessary to compute meaningful
mutational likelihood scores from matched samples, so we only performed this
analysis on the tumour types in Figure 4C.

4) The authors should clarify and justify why they focus specifically on hepatocellular
carcinoma in Figure 5, rather than analyzing other tumor types. What is observed in
other cancer types where CTNNB1 mutations are prevalent?

We thank the reviewer for pointing out the lack of clarity on this point in our original
submission. To integrate MES values with signalling in tumours, matched
genome/exome and transcriptome data are required. For most of the COSMIC primary
tumour sites shown in Figure 4D, exon 3 mutations are rare, and the large n values
come from panel sequencing studies performed on thousands of patient samples
which lack transcriptome data. Of tumour types in TCGA, Hepatocellular Carcinoma
was uniquely suitable for the analysis of inter-tumour heterogeneity for the following
reasons:

- - Exon 3 mutations are common, occurring in approximately 22% of all TCGA
samples ($n = 80$). Only endometrial carcinoma had comparable numbers of exon
3 mutant tumours ($n = 90$). The next closest tumour type, Melanoma, had just 10
relevant samples. Tumour types with small numbers of samples are generally
uninformative due to the large number of other factors that contribute to inter-
tumour heterogeneity (e.g. differences in aetiology, treatment, genetic
differences in the tumour and germline, etc.).
- - In HCC, MES values showed a natural bimodal distribution (Reviewer Figure 1),
with ~65% STRONG and 35% WEAK effect, suggesting the existence of distinct
disease types. This is supported by our discovery of immune exclusion in the
STRONG group (Figure 5F, 5G). MES values in Endometrial Carcinoma were not
bimodally distributed but skewed heavily towards STRONG.
- - HCC has very few other driver mutations present at high frequency ($n = 2$ genes
mutated in >20%) that could modify the signalling effect of exon 3 mutations. For
comparison, endometrial carcinoma has $n = 24$ genes mutated in >20% of
samples, including several chromatin modifiers.

During the revision period, we screened several additional non-TCGA cohorts for which
matched exome and transcriptome are available. In a colorectal cancer cohort (SCORT
collection, $n = 2000$ samples) we found fewer than 10 samples with exon 3 missense
mutations. In a melanoma cohort (Leeds cohort, $n = 524$ samples) we found only 5

samples with exon 3 missense mutations. We were able to identify n = 58 patients with
exon 3 mutations in a Juvenile nasopharyngeal angiofibroma cohort (Sanger cohort),
but n = 55 of these mutations were high effect.

In our revised manuscript, we justify our focus on HCC starting on line 344:

*“To gain insight into the effect of CTNNB1 mutation strength on inter-tumour heterogeneity*
*we focused on HCC, due to the high rate of exon 3 mutations and bimodal distribution of*
*MES values in this tumour type”.*

The reasons that TCGA cohorts from other β -catenin-dependent tumour types were not
informative are now summarised in the ‘Study Limitations’ section at the end of our
revised submission (line 489).

*“...other CTNNB1-dependent tumour types in TCGA were not informative either due to low*
*sample number (e.g. adrenocortical carcinoma), low frequency of exon 3 mutant tumours*
*(e.g. colorectal adenocarcinoma, melanoma) or low diversity of MES values (e.g.*
*endometrial carcinoma).”*

5) Figure callouts on p11-12 appear incorrect and should be rechecked. For example
the callout on line 332 and the callout on line 340

We are grateful to the reviewer for pointing this out. We have checked all figure call outs
in the manuscript and corrected those which were incorrect.

6) Regarding experimental validation, it would enhance the rigor of the study to also
show that individually cloned variants (or single cell clones of the screened population)
confirm the results of the sequencing-based screen. Authors could validate variants
from within each bin of CTNNB1 activity.

We agree and, as detailed in full in response to Reviewer 1 point 3, we have now
validated a total of 11 different variants in 15 clonal mESC lines, spanning a wide range
of MES values, focusing on codon positions that are most frequently mutated in human
cancer, following their introduction into the mESC genome via genome editing. The
resulting data are shown above in Reviewer Figure 9 and are included as part of Figure 2
(panels B, C & D) in the revised submission.

7) The paper presents data from multiple tumor types, then focuses on HCC. However
it is not clear if the scores from ES cells best represent the function of CTNNB1 in one
tissue type or another. The interpretation given for strong vs. weak mutations in a given
tumor type is that HCC may select for weakly activating CTNNB1 mutations. However,
it seems possible that in HCC cells the phenotypes of these mutations might be
different (for example if a CTNNB1 interacting protein is expressed in HCC but not ES or

vice versa). The conclusions of the paper would be enhanced by a counter screen in
HCC cells or other evidence that the *CTNNB1* mutations result in the same functional
groups in HCC cells.

We agree and thank the reviewer for raising this important point. However, as detailed
in response to Reviewer 1 point 2, we believe that HCC cell lines do not provide the best
background to test the function of *CTNNB1* variants because they either already have a
constitutively high level of β -catenin activation, which would likely prevent our ability to
distinguish variants of low or moderate effect, or are not Wnt-dependent and thus lack
physiological relevance. Instead, we have taken two alternative approaches to test the
relevance of our scores to HCC in settings that we consider to be more physiological.

First, we have improved our analysis of the TCGA HCC cohort and added data from a
second independent HCC cohort. In both cases, we find that mutations observed in
HCC that are called as WEAK effect in our mESC screen are associated with
significantly lower transcript levels for known β -catenin targets (Reviewer Figure 7).
Second, we report positive correlations between MES values from mESCs and β -
catenin target transcript levels in 'normal' human fetal liver organoids that have been
homozygously edited with one of four different exon 3 mutations (Reviewer Figure 8).
These data support the relevance of our mESC system for predicting *CTNNB1* variant
effects in HCC.

G. References are appropriate.

H. The manuscript writing was clear and appropriate.

Reviewer #3:

Remarks to the Author:

Krishnan et al. describe a small multiplex assay of variant effect to identify mutations in
the β -catenin degron that disrupt protein-protein interactions and lead to stabilization
of β -catenin. The authors use a modified saturation genome editing approach to
introduce mutations into an engineered mouse ESC line. Then they stratify mutations in
the degron by transcriptional activation using a standard flow-seq strategy with a β -
catenin-responsive reporter. They identify mutations in codon positions known to be
important for E3 ligase binding and phosphorylation suggesting the assay is behaving as
designed. The authors compare their data to an unsupervised variant effect predictor
and note that EVE does not do as well as the assay at identifying hotspot codons (as
one might expect). The authors find that many mutations found in cancer are bimodally
distributed between weak and strong activation of β -catenin and that there was a
difference in survival between these two populations. The authors then characterize the
phenotypic differences between these two populations and note differential gene
expression and immune cell invasion. In summary, this is a neat use of mutational
scanning to explore a long-standing question in β -catenin-driven cancer.

We are very grateful to Dr Starita for this summary and positive comments on our
manuscript, as well as the constructive suggestions for improvement.

I do have points that should be addressed before publication.
Please address the following:

The methods section is missing key information.

1)

- How many clones?
- How many cells transfected?
- What was the replicate structure? Multiple transfections or one pool of cell and replicate flow sorts?

We performed two separate transfections, each on 200×10^6 cells (400×10^6 cells total). Each set of cells went through the same editing, drug selection, and flow sorting procedure detailed in Figure 1D of our revised manuscript. This information was included in our original methods section under the heading “transfection of homology-directed repair template library and flow sorting”, and is now detailed on lines 551 – 569 of our revised methods section.

Due to the FIAU selection step in our protocol, which selectively kills cells that retain the puro Δ TK cassette (See Figure 1D of our revised submission), we are unable to say precisely how many clones were generated in each transfection. Based on prior experience we conservatively estimate transfection rates of 20%, meaning approximately 40 million cells were exposed to genome editing in each of the two transfections. Even if HDR rates were 1%, which is low for mESCs, this would yield a minimum of 400,000 clones per replicate: >1000-fold coverage of 360 repair templates.

2) How many replicate experiments were performed and over what time frame?

The replicate experiments described above were performed over a time frame of one month.

3) Scatter plots of replicates are required. Can you throw out the worst behaved replicates and improve data quality?

We have added scatter plots in Figure S3 comparing normalised read counts for each missense mutation ($n = 342$) across each of the six bins in the two experimental replicates. Normalised read counts in each GFP bin correlated well between replicates, so in our original submission, reads were combined to generate MES values. However, to address whether one replicate was of superior quality to the other, we recalculated separate MES values from replicate 1 and replicate 2 without pooling reads. The Spearman rho correlation for missense mutations across replicates is 0.662. For benchmarking, we compared each replicate with EVE predictions for the same set of 342 missense mutations and obtained Spearman rho values of 0.662 for replicate 1, 0.660 for replicate 2, and 0.663 for the original combined score. We therefore infer that the two replicates are of similar quality, and have retained our original MES values which combine data from both.

4) How many cells were sorted in each experiment? How many in each bin?

This information was provided in Table S7 of our original submission, now Table S8 in
 our revised submission, and is reproduced below (Reviewer Table 2):

Replicate 1	cells sorted	Replicate 2	cells sorted
Unsorted pool	>200,000	Unsorted pool	>200,000
bin 1	200,000	bin 1	200,000
bin 2	200,000	bin 2	200,000
bin 3	200,000	bin 3	200,000
bin 4	80,000	bin 4	1,130,000
bin 5	445,427	bin 5	942,000
bin 6	8,309	bin 6	11,641

 **Reviewer Table 2:** Cell numbers sorted into each GFP bin across two replicate experiments.

5) How much DNA (converted to genome equivalents) was used to amplify in the PCR?

All of the genomic DNA purified from the sorted cells enumerated in Reviewer Table 2
 was used to generate libraries. However, we are unable to say with confidence how
 much genomic DNA was used in each set of reactions because, regrettably, the DNA
 was quantified using a Nanodrop. Conversion of the resulting DNA quantities into
 genome equivalents in some cases yielded values that are higher than possible based
 on the number of cells sorted in the corresponding bin. For this reason, we believe that
 it is not appropriate to include DNA quantities in our revised manuscript.

While we acknowledge that information on genome equivalents would help to
 accurately assess coverage, our data strongly suggest that sampling of the 342
 mutations was more than adequate, even in bins where relatively few cells were sorted.
 The histograms now shown in Figure 1E of our revised submission (Reviewer Figure 12)
 illustrate the diversity of mutations observed in bin 6, where the lowest total number of
 cells (19,950) were sorted. In bin 6, 108/342 mutations were observed at normalised
 frequencies that exceeded those observed in the unsorted GFP pool, and this was not
 substantially different from other bins with high GFP activity and more cells (105/342 for
 bin 5, 116/342 for bin 4).

 **Reviewer Figure 12:** The frequency of individual missense mutations across each position in the
 mutation hotspot in cell populations sorted according to the scheme shown in panel D, expressed
 relative to their frequency in the unsorted pool sample. The colour scheme used to represent
 different missense mutations is shown underneath.

6) The Pearson R range for replicate experiments in the methods section does not match the range reported in the results section.

We are grateful for pointing out this discrepancy and have corrected the correlation coefficients in our resubmission. The r values have changed slightly from those reported in the methods section of our original submission because we realised that wildtype codons had been included in the original calculations, when several of these were undetectable and therefore yielded zero values in both replicates for all bins, as explained further below in response to Reviewer 3 point 8. The correct range of Pearson r values between replicates across normalised missense allele frequencies for GFP-sorted bins is 0.54 – 0.89, with 0.94 for the untransfected plasmid and 0.63 for the unsorted cell pool. Dot plots are included in Figure S3C. We apologise for this error.

7) What parameters were used for NGmerge (were low quality reads filtered?)

NGmerge does not perform quality filtering. This part of the read processing pipeline was run in two stages, first to trim overhangs leaving only the overlapping sequences of read pairs, then to merge read pairs into a single consensus sequence. This increases the quality scores because evidence from both reads contributes to each base call. Because average quality scores were high (Reviewer Figure 13), we did not consider it necessary to filter low quality reads. We have added the following text to the methods section to clarify this point.

Adapters were trimmed and paired ends merged with NGmerge

(<https://github.com/harvardinformatics/NGmerge>) to produce single end reads. Mean sequence quality (Phred) scores were high (>30) across all base positions targeted for mutagenesis, so low quality reads were not filtered out. Single reads were then aligned with bwa mem v0.7.17 (Vasimuddin et al. 2019) to the 162bp CTNNB1 reference sequence. A set of reads with a single missense on-target mutation, and no other mutations, was generated. Reads that did not fully cover the region targeted for mutagenesis (58-111bp) were excluded, as were alignments with any of: indels anywhere, no mutations in the target region, mutations only outside the target region, multiple mutations in the target region, synonymous mutations only, or mutations resulting in a codon that was not in the repair template library. Remaining reads had precisely one missense mutation specified from the HDR template library.

Reviewer Figure 13. FastQC plot showing the mean quality (Phred) scores across each base position in the merged consensus read libraries. The exon 3 hotspot region targeted for saturation mutagenesis is indicated above the plot.

8) Why were the synonymous variants filtered out? They can serve as a nice neutral control.

We are thankful for this comment and agree that synonymous variants should ideally serve as a neutral control. Our codon-level saturation donor library included one codon per amino acid. In 50% of codon positions (9/18), that codon happened to be identical to the one encoded by the genome so could not be identified based on mismatches from the consensus sequence, whereas in the remainder it was possible to enumerate relevant sequencing reads based on one or more synonymous nucleotide differences. When MES values are calculated including reads that can be assigned to these 9 synonymous changes, we were surprised to find that the synonymous mutant codons yielded a range of MES values, sometimes in the upper half of the 20 total values for the same codon position. However, we also noticed that all 9 synonymous mutant alleles were present at unusually low frequency in the untransfected donor plasmid pool (observed normalised mutant allele frequency below 0.028% versus expected normalised mutation frequency of 0.28% based on equal representation of 351 mutations (342 non-synonymous and 9 synonymous). We assume this occurred due to abnormal growth or processing of bacterial stocks before template pooling. For this reason, this control would not serve its intended purpose and so we prefer to leave the information out.

However, the observation that synonymous mutations were underrepresented prompted us to consider whether certain missense mutant alleles might be similarly under-represented in the donor template pool, and hence give similarly unreliable scores. 31 of 342 missense mutations had normalised read counts in the same range as the 9 synonymous mutations (less than 0.028% of normalised read counts in the untransfected donor plasmid pool). Of note, none of the 31 corresponded to a mutation observed in the TCGA or Montironi HCC cohorts analysed in Figure 5 of our paper, so their impact on the conclusions of our paper is negligible. Nonetheless, in our revised manuscript we have added a column to Table S2 where the full MES set is listed, so that these ‘lower confidence’ scores can be identified. This is explained in the revised methods section (lines 648 – 653).

9) “Replacement frequency” is confusing to me and not a common way of referring to mutation abundance.

We have removed this term and replaced it with ‘mutant allele frequency’.

10) Scatter plot for S3 please.

Scatter plots for plasmid and pool samples are now included as part of Figure S3 in our revised submission.

11) The framing of the comparison to EVE is off base. Of course, the assay is going to be better at identifying GOF mutations, you’ve set up an assay to select for them. The fact that EVE can identify what are essentially GOF mutations at all is impressive (Livesey

and Marsh 2023). This part of the paper should be edited to reflect that and not that the
assay won a straw-man competition vs. a computational variant effect predictor.

We completely agree that it is impressive EVE can identify gain-of-function mutations; it
was not our intention to suggest otherwise. As stated in our original submission:

“the correlation between the MES values for the *CTNNB1* mutation hotspot and the
corresponding EVE predictions (Spearman’s $\rho=0.663$) was higher than observed for any
other protein in this benchmarking study, and higher than observed for all but 3 out of 1430
VEP/DMS comparisons (Livesey and Marsh, 2023)”

This shows that EVE performs well not only in the identification of pathogenic gain-of-
function *CTNNB1* mutations relative to other variant effect predictors, but also relative
to its own predictions on other genes (although the relatively small size of our dataset
likely influences these rankings). Given this, in our original submission we thought it
was interesting to highlight the small number of codon positions at which EVE
predictions substantially diverged from our experimental values. However, because
this was a minor part of our manuscript, to avoid the perception of a straw-man
competition we have removed Figure S3 from our original submission which highlighted
the discordance at positions 38 and 44. We have also removed the survival analysis of
HCC patients stratified on the basis of EVE predictions that was shown in Figure S6 of
our original submission.

12) Also, the second paragraph that discusses EVE’s erroneous identification of two
codon positions (G38 and P44) that aren’t selected for in the assay. EVE uses multiple
sequence alignments to query evolutionary conservation of those codons. The assay in
this manuscript assesses a single function of β -catenin. It is entirely possible that
evolution has constrained these amino acids for other reasons not assessed in this
assay. Your assessment that the assay can better stratify β -catenin mutation important
for cancer may be correct, but likely not because EVE is wrong.

We agree, and our original submission did not state that EVE’s assignment of high
perturbation scores to substitutions at G38 and P44 was ‘erroneous’, simply that it was
not predictive of mutation frequencies in human cancer. Nonetheless to avoid
misunderstanding we have removed this passage of text and Figure from our revised
submission.

Line 240. The word unusual should be removed. The whole point of the functional data
y’all worked so hard to generate was to do this analysis, correct?

We have removed the word unusual from this passage of text.

The TCGA and COSMIC analyses are outside my expertise but look compelling to me

**Austinat, M., Dunsch, R., Wittekind, C., Tannapfel, A., Gebhardt, R. and Gaunitz, F.**
(2008). Correlation between β -catenin mutations and expression of Wnt-signaling target
genes in hepatocellular carcinoma. *Mol. Cancer* **7**, 21–21.

**Bakker, E. R. M., Hoekstra, E., Franken, P. F., Helvensteijn, W., Deurzen, C. H. M.**
**van, Veelen, W. van, Kuipers, E. J. and Smits, R.** (2013). β -Catenin signaling dosage
dictates tissue-specific tumor predisposition in Apc-driven cancer. *Oncogene* **32**, 4579–
4585.

**Geurts, M. H., Gandhi, S., Boretto, M. G., Akkerman, N., Derks, L. L. M., Son, G. van,**
**Celotti, M., Harshuk-Shabso, S., Peci, F., Begthel, H., et al.** (2023). One-step
generation of tumor models by base editor multiplexing in adult stem cell-derived
organoids. *Nat. Commun.* **14**, 4998.

**Johnson, J. L., Yaron, T. M., Huntsman, E. M., Kerelsky, A., Song, J., Regev, A., Lin,**
**T.-Y., Liberatore, K., Cizin, D. M., Cohen, B. M., et al.** (2023). An atlas of substrate
specificities for the human serine/threonine kinome. *Nature* **613**, 759–766.

**Kielman, M. F., Rindapää, M., Gaspar, C., Poppel, N. van, Breukel, C., Leeuwen, S.**
**van, Taketo, M. M., Roberts, S., Smits, R. and Fodde, R.** (2002). Apc modulates
embryonic stem-cell differentiation by controlling the dosage of β -catenin signaling. *Nat.*
*Genet.* **32**, 594–605.

**Montironi, C., Castet, F., Haber, P. K., Pinyol, R., Torres-Martin, M., Torrens, L.,**
**Mesropian, A., Wang, H., Puigvehi, M., Maeda, M., et al.** (2023). Inflamed and non-
inflamed classes of HCC: a revised immunogenomic classification. *Gut* **72**, 129–140.

**Pollard, S. M.** (2013). Neural Progenitor Cells, Methods and Protocols. *Methods Mol. Biol.*
**1059**, 13–24.

**Provost, E., Yamamoto, Y., Lizardi, I., Stern, J., D’Aquila, T. G., Gaynor, R. B. and**
**Rimm, D. L.** (2003). Functional Correlates of Mutations in β -Catenin Exon 3
Phosphorylation Sites*. *J. Biol. Chem.* **278**, 31781–31789.

**Rebouissou, S., Franconi, A., Calderaro, J., Letouzé, E., Imbeaud, S., Pilati, C., Nault,**
**J., Couchy, G., Laurent, A., Balabaud, C., et al.** (2016). Genotype-phenotype
correlation of CTNNB1 mutations reveals different β -catenin activity associated with liver
tumor progression. *Hepatology* **64**, 2047–2061.

**Wong, K., Bishop, J. A., Weinreb, I., Motta, M., Velasco-Herrera, M. D. C., Bellacchio,**
**E., Ferreira, I., Weyden, L. van der, Boccacino, J. M., Lauri, A., et al.** (2025). Wnt/ β -
catenin activation by mutually exclusive FBXW11 and CTNNB1 hotspot mutations drives
salivary basal cell adenoma. *Nat. Commun.* **16**, 4657.

Response to Reviewers document

We are grateful to the three reviewers for taking the time to provide comments on our revised manuscript. We respond to their points below.

Reviewer #1 (Remarks to the Author):

The authors have addressed most of the reviewer comments through additional experiments and supporting data sets. I am still rather concerned about the paucity of direct data provided in human model systems. For example, in mouse - loss of APC leads to tumours in the small intestine, not colon highlighting genetic difference in this pathway between mouse and human with consequently divergent phenotype. To this point, the relevance to human cancer is still somewhat lacking as the screens are performed in mouse centric models and human evidence is mainly correlative and inferential from human cancer or external organoid gene expression datasets. While the authors make a technical case for performing the screens in primary lines versus cancer lines, there is a missed opportunity as I believe the screens could have been performed or at least more direct functional evidence provided in human models such as organoid or primary lines.

We thank the reviewer for sharing their concern. The reasons for the small intestine tumours in Apc mouse models are more nuanced than genetic differences in the pathway between species. There is, for instance, a clear effect on the site of gastrointestinal tumours of diet (Yang et al 2008, PMID: 18794118) and specific metabolites (PMID: 30794774). But more importantly, no result that we could theoretically obtain from experiments using human 2D cancer cell lines would change the fact that data from primary untransformed mouse cells proved to be predictive in human cancer patients and human 3D organoids.

Reviewer #2 (Remarks to the Author):

The authors have done an impressive job responding to reviewer comments, in particular the mutation likelihood score correction/analysis and the validation in clonal ESC lines. Overall, all of my major concerns have been addressed.

Minor comments:

- In Figure 4C it should be stated in the legend if the wilcoxon test shown was or was not adjusted for mutation likelihood.

This information is now included in the Figure legend.

- In Figure 5E and G, authors should put the statistical test mentioned in the text in the figure and legend as well.

This information is now included in the text and figure legend.

Reviewer #3 (Remarks to the Author):

I am satisfied with the revisions.